

# Patterns in evolutionary origins of heme, chlorophyll *a* and isopentenyl diphosphate biosynthetic pathways suggest non-photosynthetic periods prior to plastid replacements in dinoflagellates

Eriko Matsuo[1] and Yuji Inagaki[1,2]

[1] Graduate School of Biological and Environmental Sciences, University of Tsukuba, Tsukuba, Ibaraki, Japan
[2] Center for Computational Sciences, University of Tsukuba, Tsukuba, Ibaraki, Japan

Corresponding author
Yuji Inagaki, yuji@ccs.tsukuba.ac.jp

## ABSTRACT

**Background:** The ancestral dinoflagellate most likely established a peridinin-containing plastid, which have been inherited in the extant photosynthetic descendants. However, kareniacean dinoflagellates and *Lepidodinium* species were known to bear "non-canonical" plastids lacking peridinin, which were established through haptophyte and green algal endosymbioses, respectively. For plastid function and maintenance, the aforementioned dinoflagellates were known to use nucleus-encoded proteins vertically inherited from the ancestral dinoflagellates (vertically inherited- or VI-type), and those acquired from non-dinoflagellate organisms (including the endosymbiont). These observations indicated that the proteomes of the non-canonical plastids derived from a haptophyte and a green alga were modified by "exogenous" genes acquired from non-dinoflagellate organisms. However, there was no systematic evaluation addressing how "exogenous" genes reshaped individual metabolic pathways localized in a non-canonical plastid.

**Results:** In this study, we surveyed transcriptomic data from two kareniacean species (*Karenia brevis* and *Karlodinium veneficum*) and *Lepidodinium chlorophorum*, and identified proteins involved in three plastid metabolic pathways synthesizing chlorophyll *a* (Chl *a*), heme and isoprene. The origins of the individual proteins of our interest were investigated, and we assessed how the three pathways were modified before and after the algal endosymbioses, which gave rise to the current non-canonical plastids. We observed a clear difference in the contribution of VI-type proteins across the three pathways. In both *Karenia*/*Karlodinium* and *Lepidodinium*, we observed a substantial contribution of VI-type proteins to the isoprene and heme biosynthesises. In sharp contrast, VI-type protein was barely detected in the Chl *a* biosynthesis in the three dinoflagellates.

**Discussion:** Pioneering works hypothesized that the ancestral kareniacean species had lost the photosynthetic activity prior to haptophyte endosymbiosis. The absence of VI-type proteins in the Chl *a* biosynthetic pathway in *Karenia* or *Karlodinium* is in good agreement with the putative non-photosynthetic nature proposed for their ancestor. The dominance of proteins with haptophyte origin in the *Karenia*/*Karlodinium* pathway suggests that their ancestor rebuilt the particular pathway by

genes acquired from the endosymbiont. Likewise, we here propose that the ancestral *Lepidodinium* likely experienced a non-photosynthetic period and discarded the entire Chl *a* biosynthetic pathway prior to the green algal endosymbiosis. Nevertheless, *Lepidodinium* rebuilt the pathway by genes transferred from phylogenetically diverse organisms, rather than the green algal endosymbiont. We explore the reasons why green algal genes were barely utilized to reconstruct the *Lepidodinium* pathway.

## INTRODUCTION

Dinoflagellates are aquatic unicellular eukaryotes belonging to one of the major taxonomic groups of eukaryotes, Alveolata. About half species of the dinoflagellates described to date are photosynthetic (*Taylor, Hoppenrath & Saldarriaga, 2008*). The vast majority of photosynthetic dinoflagellates harbor plastids containing chlorophylls *a* and *c* (Chl *a+c*), which are remnants of a red algal endosymbiont captured by the common ancestor of dinoflagellates (*Hoek, Mann & Jahns, 1995*; *Janouškovec et al., 2010*). Compared to red alga-derived, Chl *a+c*-containing plastids in other eukaryotic algae, namely cryptophytes, chromerids (e.g., *Chromera velia* and *Vitrella brassicaformis*), stramenopiles, and haptophytes, the plastids in the vast majority of dinoflagellates are unique in containing a carotenoid called peridinin (*Jeffrey, Sielicki & Haxo, 1975*; *Zapata et al., 2012*). However, some species are known to bear "non-canonical" plastids, which are distinctive from "peridinin-containing plastids" in the majority of photosynthetic dinoflagellates in both pigment composition and evolutionary origin. The plastids in members of genera *Karenia* and *Karlodinium* (family Kareniaceae) contain 19′-hexanoyl-fucoxanthin along with Chl *a+c*, which are remnants of an endosymbiotic haptophyte (*Bjørnland, Haxo & Liaaen-Jensen, 2003*; *Tengs et al., 2000*; *Zapata et al., 2012*). Members of the genus *Lepidodinium* (family Gymnodiniaceae) established the current plastids containing chlorophylls *a* and *b* through the endosymbiosis of a pedinophyte green alga (*Watanabe et al., 1987*, *1990*; *Matsumoto et al., 2011*; *Kamikawa et al., 2015a*). At the morphological level, the endosymbionts in the two lineages described above were extensively reduced in the host cells, leaving only plastids in the endosymbiont-derived compartments. At the genetic level, endosymbiont genes, particularly ones for plastid functions and maintenance (henceforth we designate as "plastid-related genes"), were transplanted into the host genomes of the two dinoflagellate lineages (*Ishida & Green, 2002*; *Takishita, Ishida & Maruyama, 2004*; *Nosenko et al., 2006*; *Patron, Waller & Keeling, 2006*; *Minge et al., 2010*; *Burki et al., 2014*). Such gene transfer from an endosymbiont to its host (endosymbiotic gene transfer or EGT) is regarded as one of the keys for the host-endosymbiont interlock at the genetic level. Thus, both haptophyte-derived and green alga-derived plastids in dinoflagellates are regarded as genuine organelles in the current host cells.

In addition to endosymbiotically transferred genes, *Karenia brevis*, *Karlodinium veneficum*, and *Lepidodinium chlorophorum* are known to possess plastid-related genes, which bear phylogenetic affinities to the orthologous genes in peridinin-containing dinoflagellates (*Takishita, Ishida & Maruyama, 2004*, *Takishita et al., 2008*; *Minge et al., 2010*; *Nosenko et al., 2006*; *Patron, Waller & Keeling, 2006*; *Burki et al., 2014*; *Bentlage et al., 2016*). As the peridinin-containing plastid was most likely established in the ancestral dinoflagellates, it is reasonable to assume that the genes described above have been inherited vertically throughout the dinoflagellate evolution, rather than acquired from the endosymbiont. Furthermore, a certain fraction of plastid-related genes was unlikely to be vertically inherited from the ancestral dinoflagellates or endosymbiotically acquired from eukaryotic algae that gave rise to non-canonical plastids (*Nosenko et al., 2006*; *Patron, Waller & Keeling, 2006*; *Minge et al., 2010*; *Burki et al., 2014*; *Bentlage et al., 2016*). In light of previous gene surveys in the *Karenia brevis/Karlodinium veneficum* and *L. chlorophorum* transcriptomic data, it is clear that the proteomes in the non-canonical plastids in these dinoflagellates are composed of the proteins with diverse evolutionary origins. However, to our knowledge, it has been unclear whether the overall degree of evolutionary "chimerism" in plastid proteome varies among non-canonical plastids established separately in dinoflagellate evolution. Likewise, we are uncertain whether there is an evolutionary consistent pattern in chimerism across multiple plastid-localized metabolic pathways within a non-canonical plastid. Prior to this study, the individual evolutionary origins of enzymes involved in the heme and isopentenyl diphosphate (IPP) biosynthetic pathways in diverse "complex algae," which acquired the plastids through green or red algal endosymbioses, have been characterized (*Wilhelm et al., 2006*; *Cihlář et al., 2016*; *Bentlage et al., 2016*), albeit their overall patterns has not been compared to each other. In this study, we investigate the evolutionary origins of enzymes involved in the biosyntheses of heme and IPP, as well as that of chlorophyll *a* (Chl *a*), which links to the two aforementioned pathways, in *Karenia brevis/Karlodinium veneficum* and *L. chlorophorum*.

The vast majority of the enzymes involved in C5 pathway for the heme biosynthesis, the non-mevalonate pathway for IPP biosynthesis, and the Chl *a* biosynthesis is nucleus-encoded in photosynthetic eukaryotes (*Oborník & Green, 2005*; *Bentlage et al., 2016*). The three pathways in photosynthetic eukaryotes and those in cyanobacteria are principally homologous to each other (*Hunter, 2007*; *Sousa et al., 2013*; *Oborník & Green, 2005*), suggesting that the pathways in photosynthetic eukaryotes can be traced back to those endosymbiotically acquired from the cyanobacterial endosymbiont that gave rise to the first plastid (primary plastid). Consistent with their cyanobacterial ancestry, the three pathways described above occur in the plastids in photosynthetic eukaryotes. In the following paragraphs, we briefly overview the three pathways.

C5 pathway for the heme biosynthesis (Fig. 1) transforms glutamyl-tRNA to protoheme through nine consecutive enzymatic steps, and the seventh and eigth steps are catalyzed by functionally homologous but evolutionarily distinct enzymes (Fig. 1)—*hemN* and *hemF* for the coproporphyrinogen oxidase (CPOX) activity (*Panek & O'Brian, 2002*),

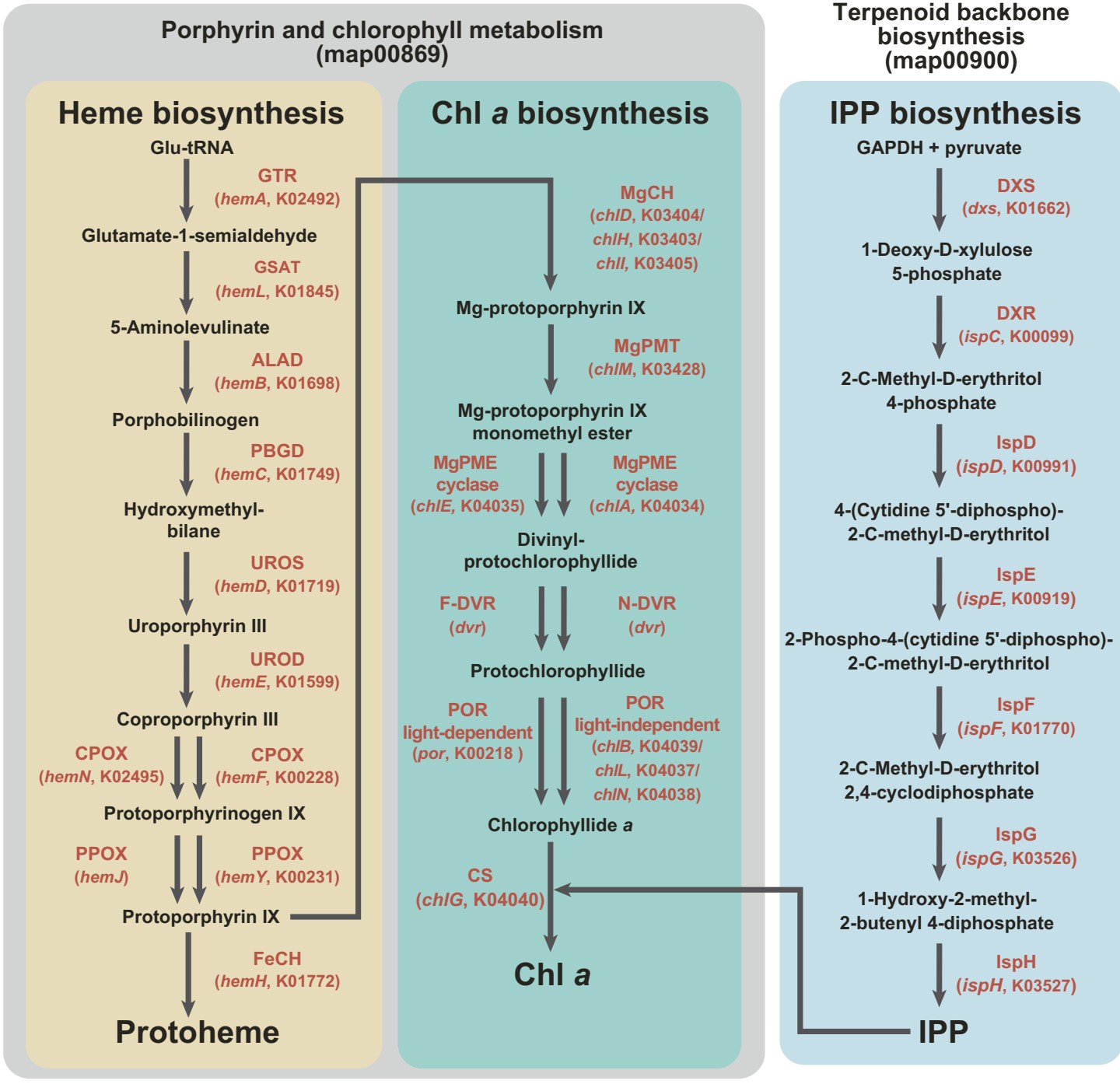

**Figure 1 Enzymes examined in this study, and their substrates and products.** C5 pathway for the heme biosynthesis is shaded in yellow. In this study, we regard the steps converting protoporphyrin IX to Chl *a* as the "Chl *a* biosynthetic pathway," and shaded in green. The non-mevalonate pathway is shaded in blue. The first two pathways belong to "Porphyrin and chlorophyll metabolism" (map00860), while the third pathway belongs to "Terpenoid backbone biosynthesis" (map00900) in Kyoto Encyclopedia of Genes and Genomes pathway (KEGG pathway, http://www.genome.jp/kegg/pathway.html). Enzymes involved in the three pathways, as well as their gene names and corresponding KOIDs, are shown in red.

and *hemJ* and *hemY* for the protoporphyrinogen IX oxidase (PPOX) activity (*Kobayashi et al., 2014*). We are aware of the heme biosynthetic pathway in apicomplexan parasites and their relatives being diversified (*Kořený et al., 2011*). However, we do not mention these exceptions below, as we can discuss the evolutions of the heme biosynthesis in *Karenia brevis/Karlodinium veneficum* and *L. chlorophorum* without acknowledging the unorthodox pathways in apicomplexan parasites and their relatives.

As shown in Fig. 1, the pathway synthesizing Chl *a* and C5 pathway for the heme biosynthesis share a common precursor, and the first to eigth enzymatic steps converting glutamyl-tRNA to protoporphyrin IX (*Reinbothe & Reinbothe, 1996*; *Beale, 1999*). We regard the steps converting protoporphyrin IX to Chl *a* as the "Chl *a* biosynthetic pathway" in this study. As seen in the heme biosynthesis, two evolutionarily distinct enzymes are known to catalyze three out of the six steps (Fig. 1). Firstly, there are two evolutionarily distinct types of Mg-protoporphyrin IX monomethyl ester cyclase (MgPME cyclase), one is a multi-subunit enzyme (*Yamanashi, Minamizaki & Fujita, 2015*; *Chen, Canniffe & Hunter, 2017*) and the other is a single-subunit enzyme encoded by *chlE* (*Yamanashi, Minamizaki & Fujita, 2015*). To our knowledge, neither of the enzymes has been detected in complex algae (e.g., diatoms; *Wilhelm et al., 2006*; *Nymark et al., 2009*). Secondly, two evolutionarily distinct types of divinyl chlorophyllide *a* 8-vinyl-reductase (DVR), namely N-DVR and F-DVR, are known, and the former and the latter use NADPH and reduced ferredoxin for electron donors, respectively (*Ito et al., 2008*; *Ito & Tanaka, 2014*). Thirdly, light-dependent and light-independent types of protochlorophyllide reductase (POR) have been identified (*Fujita & Bauer, 2003*). The light-dependent POR is a single polypeptide and encoded by a nuclear gene, *por*. The light-independent version comprises three hetero subunits encoded by *chlB*, *chlL* and *chlN*, and some/all of the three genes are plastid-encoded in photosynthetic eukaryotes.

The non-mevalonate pathway for the IPP biosynthesis comprises seven consecutive enzymatic steps (Fig. 1). To our knowledge, no functionally homologous but evolutionarily distinct enzyme has been known in this pathway. IPP is an essential precursor for various terpenoids including the phytol residue in Chl *a*. Thus, in land plants, the non-mevalonate pathway appeared to supply IPP to the Chl *a* synthesis, suggesting that the two pathways tightly couple in the plastids (*Lichtenthaler et al., 1997*; *Dubey, Bhalla & Luthra, 2003*).

We here surveyed the transcripts encoding enzymes involved in three plastid-localized metabolic pathways in two kareniacean species (*Karenia brevis* and *Karlodinium veneficum*) and *L. chlorophorum*, which bear haptophyte-derived and green alga-derived plastids, respectively. Individual proteins identified in this study were then subjected to phylogenetic analyses to evaluate how the three pathways were modified during the haptophyte/green algal endosymbiosis. Our systematic assessment revealed that the impact of EGT was different among the three pathways in the three species examined in this study. We here propose biological reasons why the impact of EGT varied among the three pathways in the dinoflagellates bearing non-canonical plastids.

## MATERIALS AND METHODS

### RNA-seq analysis of *L. chlorophorum*

*Lepidodinium chlorophorum* NIES-1868 has been maintained in the laboratory as described in *Kamikawa et al. (2015a)*. Total RNA was isolated from the harvested cells with TRIzol Reagent (Thermo Fisher Scientific, Waltham, MA, USA) according to the manufacture's instruction. We sent the RNA sample to Hokkaido System Science (Sapporo, Japan) for library construction with Truseq RNA Sample Prep Kit (Illumina, San Diego, CA, USA) followed by sequencing on a Illumina HiSeq 2000 platform. We yielded ~386 million paired-end reads (~39 Mbp in total). After exclusion of poorly called reads and removal of an adaptor sequence by FASTX-toolkit (http://hannonlab.cshl.edu/fastx_toolkit/download.html), we finally assembled the curated reads into 77,841 contigs by Trinity ver.2.1.0 (*Haas et al., 2013*). The RNA-seq data of *L. chlorophorum* was deposited in DDBJ Sequence Read Archive (DRA accession number DRA006544).

We evaluated the coverage of the contig data yielded from the *L. chlorophorum* RNA-seq data against the "universally conserved genes" in eukaryotes with BUSCO ver. 3 (*Waterhouse et al., 2017*). The proportions of "complete," "fragmented" and "missing" BUSCOs were 84.2, 3.3 and 12.5%, respectively.

### Survey of the genes encoding proteins involved in the heme, Chl *a* and IPP biosyntheses

In this study, we conducted phylogenetic analyses on the proteins involved in "Porphyrin and chlorophyll metabolism" (map00860) and "Terpenoid backbone biosynthesis" (map00900) in Kyoto Encyclopedia of Genes and Genomes pathway (KEGG pathway, http://www.genome.jp/kegg/pathway.html). KEGG Orthology (KO) identifiers (KOIDs) of the proteins subjected to the investigation in this study were K02492, K01845, K01698, K01749, K01719, K01599, K00228, K02495, K00231, K01772, K03403, K03404, K03405, K03428, K04034, K04035, K04040, K00218, K04037, K04038, K04039, K01662, K00099, K00991, K00919, K01770, K03526 and K03527 (see also Fig. 1). To generate amino acid sequence alignments covering phylogenetically diverse eukaryotic algae and prokaryotes, we surveyed the sequences of interest in the contigs generated from the RNA-seq data of *L. chlorophorum* (see above) and public databases including the Marine Microbial Eukaryote Transcriptome Sequencing Project (MMETSP; http://marinemicroeukaryotes.org), which provides the contigs from the combined assemblies of the *Karenia brevis* and *Karlodinium veneficum* RNA-seq data ("Karenia-brevis-Wilson," "Karenia-brevis-CCMP2229" and "Karlodinium-micrum-CCMP2283"; https://www.imicrobe.us), by TBLASTN or BLASTP. In the first BLAST search, multiple sequences registered in the KOIDs listed above were used as queries except the enzymes described below. Two evolutionarily distinct versions were known for PPOX ("*hemY*-type" and "*hemJ*-type"), and we used KOID K00231 as the query to identify the former type of PPOX. We surveyed *hemJ*-type PPOX sequences, which is not registered in the current KEGG pathway, by using UPF0093 membrane protein encoded by *slr1790* in *Synechocystis* sp. strain PCC 6803 (UniProtKB P72793) as the query. Both N-DVR and

F-DVR are included in a single KOID (K19073), but the sequences deposited in the GenBank database were subjected to the initial survey as queries by referring previously published works (*Meguro et al., 2011*; *Nagata et al., 2005*). The accession numbers for the queries used in the initial BLAST searches are summarized in Table S1. The candidate sequences matched with the queries with *E*-values smaller than $10^{-20}$ in the first BLAST searches were retained. These candidate (amino acid) sequences were then subjected to BLASTP searches against the NCBI nr database (threshold was set as *E*-value of $10^{-5}$). Based on the results of the second BLAST searches, we retrieved the sequences matched to the proteins involved in the heme, Chl *a* and IPP biosyntheses.

We further subjected the *Karenia brevis* and *Karlodinium veneficum* sequences selected by the procedure described above to BLASTP against the entire MMETSP database (https://www.imicrobe.us/#/projects/104) to examine potential cross-contamination among multiple RNA-seq data deposited in the database (*Marron et al., 2016*; *Dorrell et al., 2017*). The candidate sequences did not match with any sequence derived from organisms rather than *Karenia brevis* or *Karlodinium veneficum* (data not shown). Thus, we conclude that the *Karenia brevis* and *Karlodinium veneficum* sequences retrieved from the MMETSP database are highly unlikely contaminants from non-kareniacean organisms.

The nucleotide sequences (and their detailed information) identified in this study are summarized in Table S2. The conceptional amino acid sequences are available from the Dryad Digital Repository: DOI 10.5061/dryad.591sm73.

## Phylogenetic analyses

In this study, we investigated the origins of individual proteins involved in the three pathways for the heme, Chl *a* and IPP biosyntheses in *Karenia brevis*, *Karlodinium veneficum* and *L. chlorophorum* by phylogenetic analyses with the maximum-likelihood (ML) method. For each protein of interest, we aligned the amino acid sequences retrieved from the public sequence databases described in the previous section by MAFFT v7.149b (*Katoh & Standley, 2013*) with the L-INS-i option. The accession numbers of the sequences included in the alignments were listed in Table S3. The resultant alignments were manually refined, followed by exclusion of ambiguously aligned positions (see Table S4 for the details). The final alignments were individually subjected to RAxML 8.0.20 (*Stamatakis, 2014*). The ML tree was selected from 10 heuristic tree searches, each of which was initiated from a randomized stepwise addition parsimony tree. The most appropriate amino acid substitution model was selected for each alignment by ProtTest 3.4 (*Darriba et al., 2011*). The substitution models applied to the ML phylogenetic analyses are summarized in Table S4. ML bootstrap values (MLBPs) were calculated by summarizing 100 trees, each of which was inferred from a bootstrap data by a single heuristic tree search (see above). Bayesian analyses were conducted with PhyloBayes 4.1c under the CAT-GTR model with a discrete $\Gamma$ distribution with four categories (*Lartillot & Philippe, 2004*). Four independent Markov Chain Monte Carlo chains were run in parallel. The detailed settings of the PhyloBayes analyses are summarized in

Table S4. The individual alignments and Newick format treefiles inferred from both ML and Bayesian methods are available in the online repository (see above).

### In silico examination of plastid-localizing potential of N-terminal extensions

Nucleus-encoded proteins localized to complex plastids usually bear "bipartite" plastid-localizing signals, which are composed of an endoplasmic reticulum-targeting signal peptide (SP) and transit peptide-like sequence at their N-termini (*Bolte et al., 2009*). Firstly, we identified extra amino acid residues at their N-termini (N-terminal extensions) in the putative *Karenia brevis*/*Karlodinium veneficum* and *L. chlorophorum* proteins involved in the heme, Chl *a* and IPP biosyntheses by comparing the corresponding bacterial homologues (most of them are from cyanobacteria). We subsequently searched the SP in each N-terminal extension by SignalP v.4.1 (*Petersen et al., 2011*). In case of the SP being predicted, the transit peptide (TP) potential in the non-SP portions of the N-terminal extension was examined by ChloroP v.1.1 (*Emanuelsson, Nielsen & Von Heijne, 1999*). SignalP and ChloroP analyses were done with the default settings. The results from the analyses on the N-terminal extensions can be found in Table S5.

## RESULTS

Henceforth here, we designated (i) the proteins inherited from the ancestral dinoflagellate as "vertically inherited-type" or "VI-type," (ii) those acquired from an endosymbiont (i.e., haptophyte and green alga, in *Karenia brevis*/*Karlodinium veneficum* and *L. chlorophorum*, respectively) as "endosymbiotically acquired-type" or "EA-type," and (iii) those acquired from organisms distantly related to the host or endosymbiont lineages as "laterally acquired-type" or "LA-type." A certain fraction of the proteins investigated could not be categorized into any of the three types described above due to the lack of phylogenetic signal in alignments (see below).

### Heme biosynthetic pathway

We successfully identified all or most of the enzymes required for the heme biosynthesis in the transcriptomic data of *Karenia brevis*, *Karlodinium veneficum* and *L. chlorophorum*. In *Karenia brevis* and *Karlodinium veneficum*, eight out of the nine enzymes were identified—only uroporphyrinogen III synthase (UROS) and CPOX were missed in the former and latter, respectively. It is most likely that the UROS/CPOX transcripts were simply missed from the *Karenia brevis*/*Karlodinium veneficum* cDNA libraries, as UROS is essential to convert hydroxymethylbilane to uroporphyrinogen III, and CPOX is indispensable to yield protoporphyrinogen IX from coproporphyrinogen III. Fortunately, the *Karenia brevis* UROS sequence has already deposited in the GenBank database under the accession number CO063310, and was phylogenetically analyzed in this study (see below). Fifty-four out of the 60 transcripts (those encoding putative cytosolic proteins were excluded; see below) investigated here were found to bear N-terminal extensions. The SP were predicted in 21 out of the 54 N-terminal extensions. Seven N-terminal extensions
were predicted to bear both SP and TP. The details of the N-terminal extensions are summarized in Table S5.

### Proteins with evolutionarily diverse origins comprise the Karenia brevis and Karlodinium veneficum pathways

To convert glutamyl-tRNA to glutamate-1-semialdehyde, *Karenia brevis* and *Karlodinium veneficum* possesses two versions of glutamyl-tRNA reductase (GTR). We consider "*Karenia*-1" and "*Karlodinium*-1" sequences as VI-type, as they were nested within a clade with those of peridinin-containing dinoflagellates, and the clade as a whole received full statistical support in the GTR phylogeny (Fig. 2A). On the other hand, the second version of GTR in the two kareniacean species (*Karenia*-2 and *Karlodinium*-2) were most likely acquired from the haptophyte endosymbiont (i.e., EA-type). The two GTR sequences were nested within the haptophyte clade, and the haptophyte clade (including the *Karenia*-2 and *Karlodinium*-2 sequences) was supported by a MLBP of 66% and a BPP of 0.99 (Fig. 2A).

Aminolevulinic acid (ALA) is synthesized from glutamate-1-semialdehyde by glutamate-1-semialdehyde 2,1-aminomutase (GSAT). We identified a single version of GSAT in both *Karenia brevis* and *Karlodinium veneficum*. The possibility of the two GSAT sequences being acquired from the haptophyte endosymbiont can be excluded, as the haptophyte sequences (except that of *Pavlova* sp.) formed a robust clade and excluded the kareniacean sequences in the GSAT phylogeny (Fig. 2B). Nevertheless, the phylogenetic origin of either *Karlodinium veneficum* or *Karenia brevis* GSAT sequence could not be pinpointed any further. The GSAT sequences of *K. brevis* and peridinin-containing dinoflagellates grouped together in the both ML and Bayesian phylogenies, but their monophyly was poorly supported (MLBP of 8% and BPP < 0.50). The *Karlodinium veneficum* GSAT sequence showed no specific affinity to any sequence examined here. Thus, we withhold to conclude the precise origins of the GSAT sequences of *Karenia brevis* and *Karlodinium veneficum* in this study.

Synthesis of porphobilinogen from ALA is likely catalyzed by a single aminolevulinic acid dehydratase (ALAD) homologue in both *Karenia brevis* and *Karlodinium veneficum*. Both *Karenia brevis* and *Karlodinium veneficum* ALAD sequences formed a robustly supported clade with those of peridinin-containing dinoflagellates (MLBP of 99% and BPP of 1.0; Fig. 2C), suggesting that the two sequences are of VI-type.

We identified two versions of porphobilinogen deaminase (PBGD), which deaminates porphobilinogen to synthesize hydroxymethylbilane, in *Karenia brevis* (*Karenia*-1 and 2). The PBGD phylogeny (Fig. 2D) recovered (i) a clade of the sequences of peridinin-containing dinoflagellates and *Karenia*-1 sequence with a MLBP of 100% and a BPP of 0.99, and (ii) a clade of the haptophyte and *Karenia*-2 sequences with a MLBP of 73% and a BPP of 0.83. Thus, *K. brevis* seemingly uses both VI-type and EA-type enzymes for hydroxymethylbilane synthesis. We identified four PBGD sequences in *K. veneficum* (*Karlodinium*-1-4), which clearly share a single ancestral sequence. The *Karlodinium veneficum* clade appeared to be distant from the clade comprising the sequences of peridinin-containing dinoflagellates and the *Karenia*-1 sequence, suggesting that the

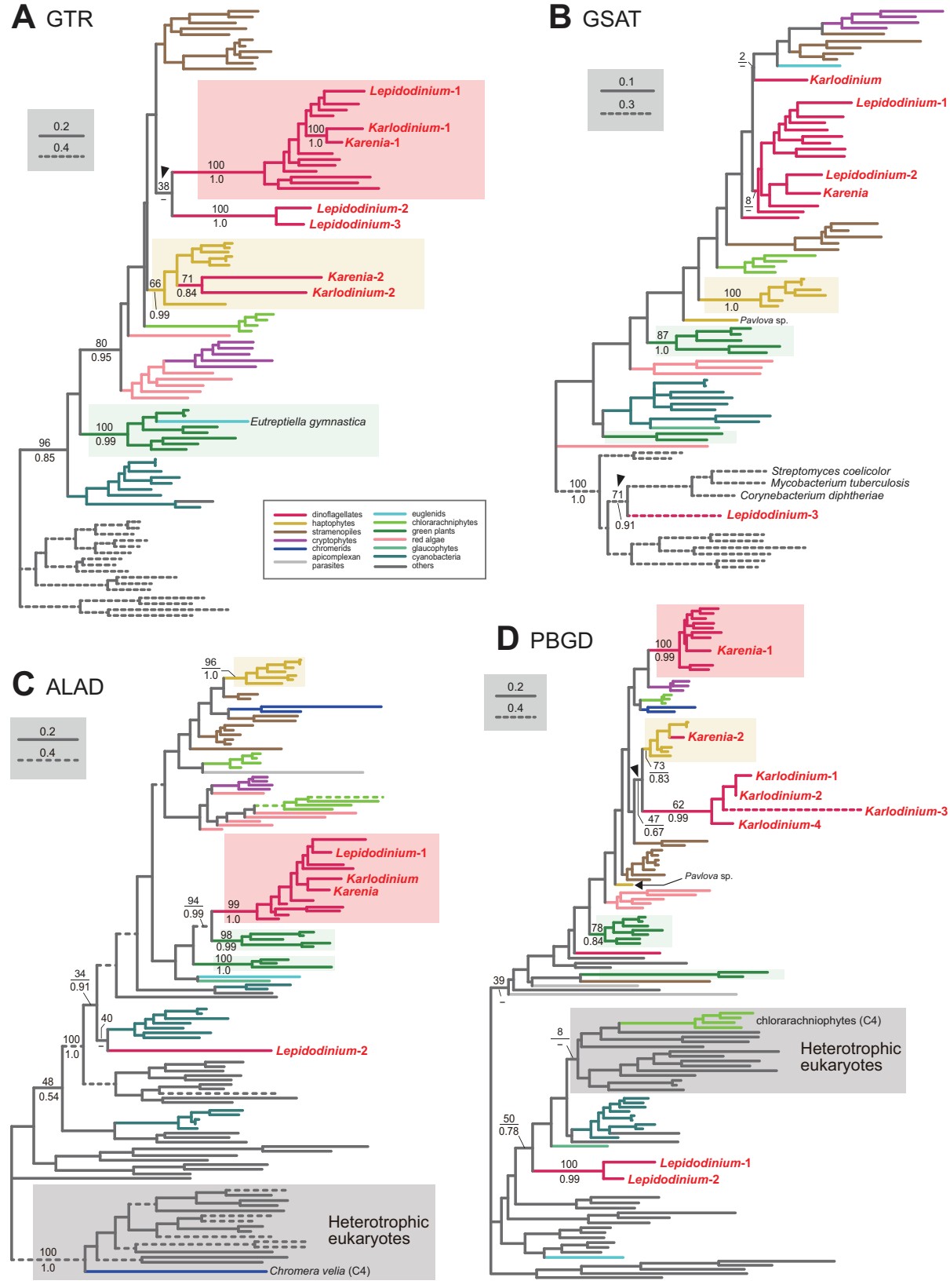

**Figure 2** **Maximum-likelihood phylogenies of 4 proteins involved in the C5 pathway for the heme biosynthesis** We provide the maximum-likelihood bootstrap values (MLBPs), as well as Bayesian posterior probabilities (BPPs), only for the selected nodes, which are important to infer the origins of the proteins of *Karenia brevis*, *Karlodinium veneficum* and *L. chlorophorum*. Dash marks represent the corresponding BPPs < 0.50. MLBPs and BPPs are shown above and beneath the corresponding nodes, respectively. Subtrees/branches are color-coded (see the inset for the details). Statistically supported clades of the dinoflagellate, haptophyte and green plant sequences are highlighted by red, yellow and green backgrounds, respectively. Clades comprising the sequences of heterotrophic eukaryotes, which were predicted to be involved in C4 pathway, were shaded in gray. (A) glutamyl-tRNA reductase (GTR). (B) glutamate-1-semialdehyde 2,1-aminomutase (GSAT). (C) delta-aminolevulinic acid dehydratase (ALAD). (D) porphobilinogen deaminase (PBGD). The identical ML trees with full sequence names and MLBPs ≥ 50% are provided as the Supplementary Materials. The substitution rates of the dashed branches are different from that of the solid lines as indicated in each figure.

*Karlodinium*-1-4 sequences are not of VI-type. The *K. veneficum* clade was connected to the haptophyte clade (including the *Karenia*-2 sequence) with a MLBP of 47% and a BPP of 0.67 (highlighted by an arrowhead in Fig. 2D). The statistical support for the particular node is insufficient to conclude or exclude the haptophyte origin of the *Karlodinium* sequences with confidence. Thus, we determined to leave the origin of the *Karlodinium* sequences uncertain.

We assessed the evolutionary origins of two distinct versions of UROS in *Karlodinium veneficum* (*Karlodinium*-1 and 2), and a single UROS sequence detected in an expressed-sequence tag data from *Karenia brevis* (Fig. 3A). The *Karenia* sequence, *Karlodinium*-1 and peridinin-containing dinoflagellate sequences grouped together with a MLBP of 76% and a BPP of 0.95. Thus, the *Karenia* and *Karlodinium*-1 sequences are most likely of VI-type. The *Karlodinium*-2 appeared to be remote from the dinoflagellate or haptophyte clade, suggesting this sequence was of LA-type (Fig. 3A). The UROS phylogeny connected the *Karlodinium*-2 sequence and that of a red alga *Rhodosorus marinus* with a MLBP of 95% and a BPP of 0.93, and the "*Karlodinium + Rhodosorus*" clade was nested within the bacterial sequences. The particular clade could have been generated by two separate gene transfers from a single bacterium (or two closely related bacteria) to *K. veneficum* and *Rhodosorus*. Alternatively, the combination of the first gene transfer from a bacterium to either *K. veneficum* or *Rhodosorus*, and the second one between the two eukaryotes may have produced the "*Karlodinium + Rhodosorus*" clade in the UROS phylogeny. In either of the two scenarios, we can regard the *Karlodinium*-2 sequence as LA-type. We need to increase the number of red algal UROS sequences in the future studies to retrace the precise origins of the *Karlodinium* and *Rhodosorus* sequences.

Four acetyl side chains in uroporphyrinogen III are removed by uroporphyrinogen decarboxylase (UROD) to generate coproporphyrinogen III. Pioneering studies revealed that photosynthetic eukaryotes with complex plastids possess evolutionarily distinct, multiple versions of UROD (*Kořený et al., 2011*; *Cihlář et al., 2016*). The UROD sequences of peridinin-containing dinoflagellates were split into three clades in the UROD phylogeny (designated as D1, D2 and D3 clades in Fig. 3B), suggesting that the three distinct versions have already been established in the ancestral dinoflagellate. Likewise, haptophytes were found to possess three distinct versions of UROD (designated as H1, H2 and H3 clades in Fig. 3B). We here identified five and four UROD sequences in *Karenia brevis* and *Karlodinium veneficum*, respectively. Among the five sequences

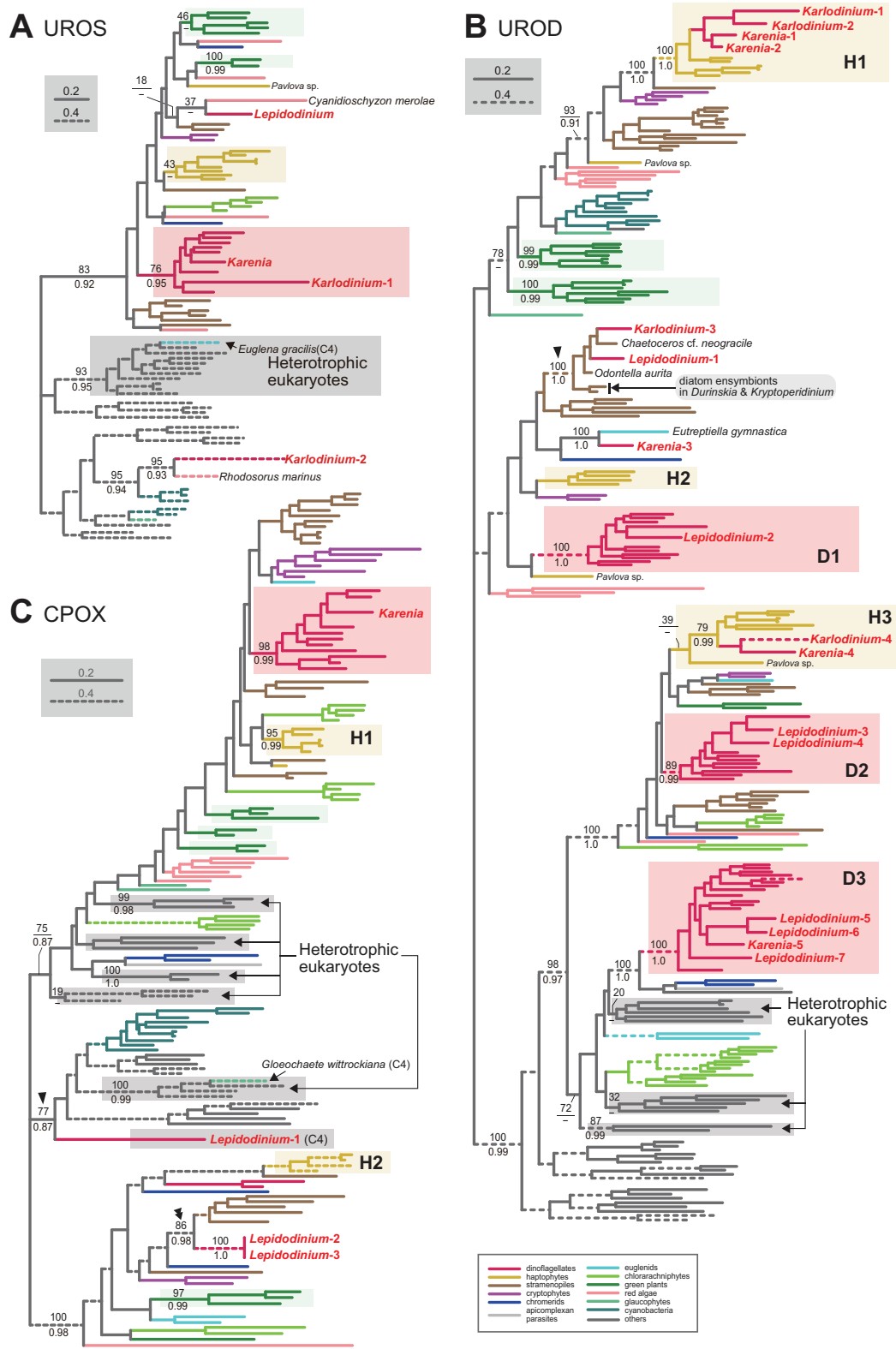

**Figure 3 Maximum-likelihood phylogenies of three proteins involved in C5 pathway for the heme biosynthesis** The details of this figure are same as those of Fig. 2. (A) uroporphyrinogen III synthase (UROS). (B) uroporphyrinogen decarboxylase (UROD). (C) coproporphyrinogen oxidase (CPOX).

identified in *K. brevis*, the "*Karenia*-1, 2 and 4" sequences were considered as EA-type, as they were placed within the haptophyte sequences in the UROD phylogeny (Fig. 3B). H1 clade including the *Karenia*-1 and 2 sequences received a MLBP of 100% and a BPP 1.0. The *Karenia*-4 sequence and the haptophyte sequences (except that of *Pavlova*) formed H3 clade, of which monophyly was supported by a MLBP of 79% and a BPP of 0.99. The "*Karenia*-3" sequence grouped with the sequence of a euglenid *Eutreptiella gymnastica* with a MLBP of 100% and a BPP of 0.99. As the *Karenia*-3 sequence showed no affinity to the dinoflagellate or haptophyte clade in the UROD phylogeny, we categorize it in LA-type. The intimate phylogenetic affinity between the *Karenia*-3 and *E. gymnastica* sequences hints either (i) gene transfer between the two organisms or (ii) two separate gene transfers from an as-yet-unknown organism to *K. brevis* and *E. gymnastica*. It is also important to improve the sequence sampling from euglenids in future phylogenetic studies to understand precisely the biological event generated the union of the *Karenia*-3 and *E. gymnastica* sequences. The "*Karenia*-5" sequence is most likely descended from one of the UROD versions established in the ancestral dinoflagellate (i.e., VI-type), as this sequence participated in D3 clade, of which monophyly received a MLBP of 100% and a BPP of 1.0. We also assessed the origins of four *K. veneficum* sequences (*Karlodinium*-1-4) based on the UROD phylogeny (Fig. 3B). The *Karlodinium*-1 and 2 sequences were nested within H1 clade, and the *Karlodinium*-4 sequence were placed within H3 clade. Thus, we conclude that the three UROD sequences are of EA-type. The *Karlodinium*-3, one of the *L. chlorophorum* sequences (see below), and diatom sequences formed a robust clade (MLBP of 100% and BPP of 1.0; highlighted by an arrowhead in Fig. 3B), suggesting that the *Karlodinium*-3 sequence were laterally acquired from a diatom (i.e., LA-type).

Coproporphyrinogen III is oxidized by CPOX to yield protoporphyrinogen IX. We identified a single version of CPOX in *Karenia brevis*, but none in *Karlodinium veneficum*. The CPOX phylogeny (Fig. 3C) recovered a clade comprising the sequences of peridinin-containing dinoflagellates and *K. brevis* with a MLBP of 98% and a BPP of 0.99. Thus, *K. brevis* uses a VI-type CPOX.

Protoporphyrinogen IX is further oxidized by PPOX to obtain protoporphyrin IX. The PPOX sequences of peridinin-containing dinoflagellates were separated into two distinct clades labeled as "D1" and "D2" (Fig. 4A), and both received MLBPs of 100% and BPPs $\geq$ 0.97. The PPOX sequences in D2 clade are likely cytosolic version, as these dinoflagellate sequences, as well as those of other photosynthetic eukaryotes bearing complex plastids (chlorarachniophytes, *Euglena gracilis* and *V. brassicaformis*), formed a robust clade with the PPOX sequences of heterotrophic eukaryotes (MLBP of 93% and BPP of 0.99; highlighted by an arrowhead in Fig. 4A). We identified two versions of PPOX in *Karenia brevis* (*Karenia*-1 and 2), while a single version was identified in *Karlodinium veneficum*. The *Karenia*-1 sequence fell into D1 clade (Fig. 4A), suggesting that this PPOX sequence is of VI-type. On the other hand, the PPOX phylogeny united the *Karenia*-2 and a single PPOX of *Karlodinium veneficum* with the sequences of stramenopiles, haptophytes and *L. chlorophorum* with a MLBP of 100% and a BPP of 0.99 (highlighted by a double-arrowhead in Fig. 4A). As the bipartitions within the clade

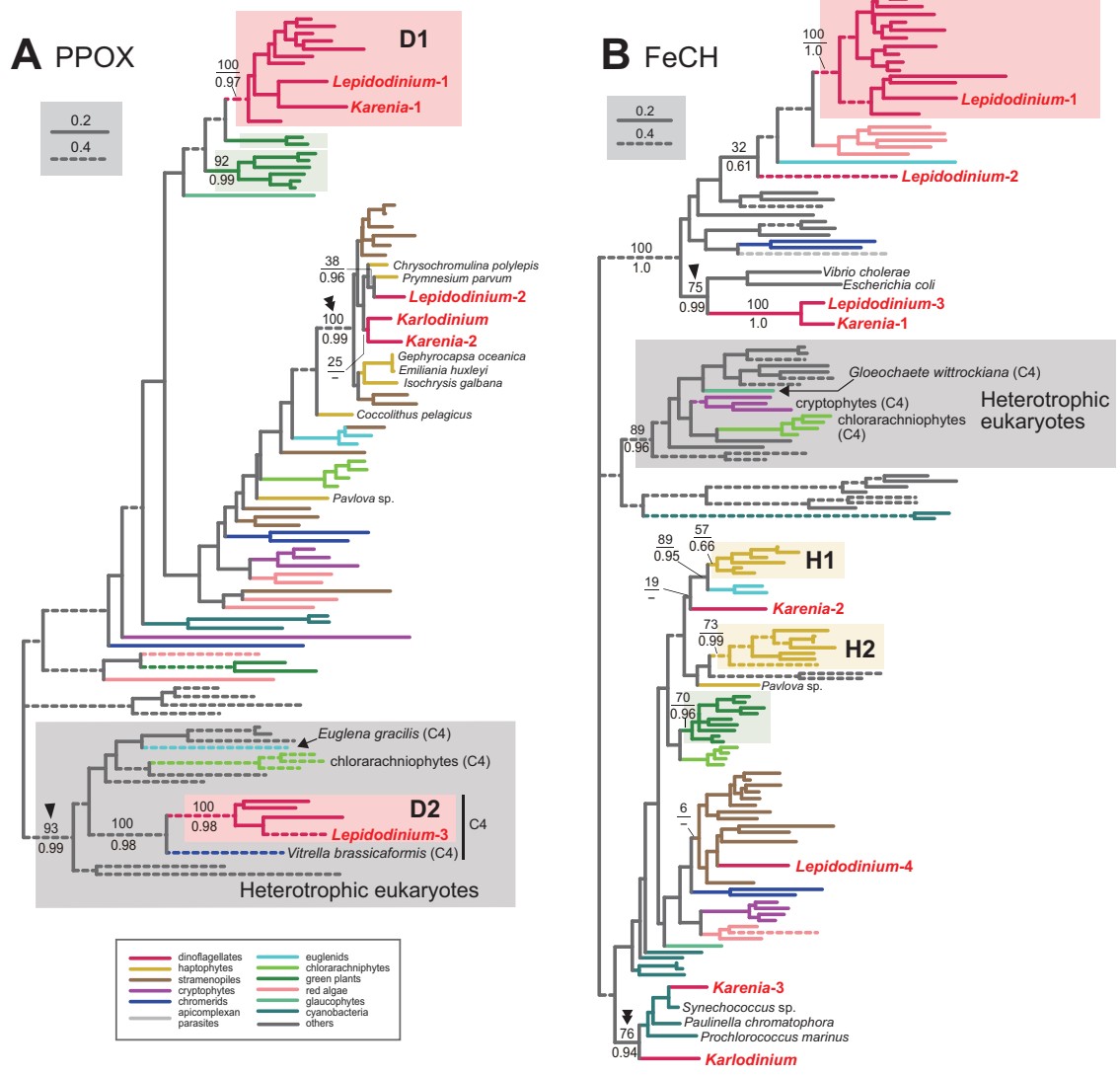

**Figure 4 Maximum-likelihood phylogenies of two proteins involved in C5 pathway for the heme biosynthesis.** The details of this figure are same as those of Fig. 2. (A) protoporphyrinogen IX oxidase (PPOX) and (B) ferrochelatase (FeCH).

were principally unresolved (Fig. 4A), we cannot exclude the possibility of the *Karenia*-2 and *Karlodinium*-1 sequences group with the haptophyte sequences in this clade. The *Karenia*-2 and *Karlodinium*-1 sequences are definitely not of VI-type, but the phylogenetic resolution was not sufficient to classify the two sequences into of EA-type or LA-type. Thus, we decide to leave the origins of the *Karenia*-2 and *Karlodinium*-1 sequences uncertain.

In the last step in the heme biosynthesis, ferrochelatase (FeCH) converts protoporphyrin IX to protoheme. None of the FeCH sequences identified in *Karenia brevis* and *Karlodinium veneficum* appeared to be of VI-type. *K. brevis* possesses three distinct versions of FeCH (*Karenia*-1-3). The FeCH phylogeny (Fig. 4B) united the *Karenia*-1 sequence with one of four versions of FeCH in *L. chlorophorum* (*Lepidodinium*-3) with a

MLBP of 100% and a BPP of 1.0, and this union was then connected specifically to γ-proteobacterial sequences with a MLBP of 75% and a BPP of 0.99 (highlighted by an arrowhead in Fig. 4B). This subtree can be explained by two sequential gene transfer events, namely the first gene transfer from a γ-proteobacterium to either *L. chlorophorum* or *K. brevis*, and the second one between the two dinoflagellates. Consequently, the *Karenia*-1 and *Lepidodinium*-3 sequences can be of LA-type, and traced back to the bacterial sequence. The *Karenia*-2 sequence is unlikely to be of VI-type, as the sequences of peridinin-containing dinoflagellates (and one of the sequences of *L. chlorophorum*) were united with a MLBP of 100% and a BPP of 1.0 (Fig. 4B). The *Karenia*-2 sequence, which showed no clear affinity to the haptophyte sequences, is unlikely to be of EA-type (Fig. 4B). Thus, we conclude that the *Karenia*-2 sequence is of LA-type, although the precise donor remains uncertain. Finally, the FeCH phylogeny grouped the *Karenia*-3 sequence and a single FeCH sequence identified in *Karlodinium veneficum* with the cyanobacterial sequences, and their monophyly was supported by a MLBP of 76% and a BPP of 0.94 (highlighted by a double-arrowhead in Fig. 4B). We conclude that the two dinoflagellate sequences are of cyanobacterial origin (i.e., LA-type).

Pioneering studies on kareniacean dinoflagellates revealed the evolutionary chimeric natures of the plastid proteomes of *Karenia brevis* and *Karlodinium veneficum*, which comprises VI-, EA- and LA-type proteins (*Ishida & Green, 2002*; *Nosenko et al., 2006*; *Patron, Waller & Keeling, 2006*; *Burki et al., 2014*). This study further enables us to evaluate the precise contribution of EA-type proteins on the evolution of the heme biosynthetic pathway in kareniacean dinoflagellates—EA-type proteins found only in three out of the nine steps, namely: (i) GTR in *Karenia brevis* and *Karlodinium veneficum*, (ii) PBGD in *Karlodinium veneficum*, and (iii) UROD in *Karenia brevis* and *Karlodinium veneficum*.

### Little impact of endosymbiotic gene transfer on the L. chlorophorum *pathway*

We identified three versions of GTR in *L. chlorophorum* (*Lepidodinium*-1-3). The *Lepidodinium*-1 sequence grouped with the sequence of peridinin-containing dinoflagellates (and the *Karenia*-1 and *Karlodinium*-1 sequences), and this "dinoflagellate" clade received a MLBP of 100% and a BPP of 1.0 (Fig. 2A). Thus, we conclude that the *Lepidodinium*-1 sequence is of VI-type. The *Lepidodinium*-2 and 3 sequences were tied to each other with a MLBP of 100% and a BPP of 1.0, and this clade was connected with the "dinoflagellate" clade described above (Fig. 2A). However, the phylogenetic affinity between the two clades received little statistical support (MLBP of 38% and BPP < 0.50; highlighted by an arrowhead in Fig. 2A), suggesting that the *Lepidodinium*-2 and 3 sequences were unlikely of VI-type. The GTR sequences of land plants and green algae (plus a euglenid *E. gymnastica*) formed a clade supported by a MLBP of 100% and a BPP of 0.99, and appeared to be distant from the two sequences of *L. chlorophorum* (Fig. 2A). Thus, we can exclude the green algal (endosymbiont) origin of the *Lepidodinium*-2 and 3 sequences. Combined, the two sequences are regarded as LA-type, although the donor of the GTR gene to *L. chlorophorum* remains unclear.

We failed to classify two out of the three versions of GSAT identified in *L. chlorophorum*. In the GSAT phylogeny (Fig. 2B), the "*Lepidodinium*-1" and

"*Lepidodinium*-2" sequences fell separately into the cluster of the sequences of peridinin-containing dinoflagellates and *K. brevis*, but this clade as a whole received no significant statistical support. Thus, we left the origins of the *Lepidodinium*-1 and 2 sequences uncertain in this study. On the other hand, the "*Lepidodinium*-3" sequence was excluded from the clade comprising the sequences of diverse photosynthetic eukaryotes and cyanobacteria with a MLBP of 100% and a BPP of 1.0, and connected to the sequences of *Streptomyces coelicolor*, *Mycobacterium tuberculosis* and *Corynebacterium diphtheriae* with a MLBP of 71% and a BPP of 0.91 (highlighted by an arrowhead in Fig. 2B). This tree topology suggests that the *Lepidodinium*-3 sequence was acquired from a bacterium (i.e., LA-type), albeit we cannot pinpoint the bacterium donated a GSAT gene to *L. chlorophorum*.

Two versions of ALAD were identified in *L. chlorophorum* (*Lepidodinium*-1 and 2). The sequences of peridinin-containing dinoflagellates, the *Lepidodinium*-1 sequence and the sequences of *Karenia brevis* and *Karlodinium veneficum* clustered with a MLBP of 99% and a BPP of 1.0 in the ALAD phylogeny (Fig. 2C). Thus, we conclude the *Lepidodinium*-1 sequence as VI-type. The *Lepidodinium*-2 sequence was distantly related to the sequences of peridinin-containing dinoflagellate or green algae/land plants, but showed no strong affinity to any clade/sequence in the ALAD phylogeny (Fig. 2C). Thus, we propose the *Lepidodinium*-2 sequence as LA-type, albeit its precise origin was unresolved in the ALAD phylogeny.

Two versions of PBGD and a single version of UROS identified from *L. chlorophorum*. The PBGD phylogeny (Fig. 2D) united the two sequences of *L. chlorophorum* together with a MLBP of 100% and a BPP of 0.99, but this clade showed little phylogenetic affinity to the sequences of peridinin-containing dinoflagellates, green algae/land plants or any sequences considered in the phylogenetic analysis. Likewise, the UROS phylogeny (Fig. 3A) recovered no clear affinity of the sequence of *L. chlorophorum* to other sequences including those of peridinin-containing dinoflagellates or green algae/land plants. Thus, *L. chlorophorum* likely uses LA-type PBGD and UROS, but their precise origins remain uncertain.

We identified seven versions of UROD in *L. chlorophorum*, and six of them showed clear affinities to the sequences of peridinin-containing dinoflagellates. In the UROD phylogeny (Fig. 3B), the sequences of peridinin-containing dinoflagellates formed three distinct clades (D1-3 clades), and each of these clades enclosed at least one sequence of *L. chlorophorum*, namely: (i) the "*Lepidodinium*-2" sequence in D1 clade, (ii) "*Lepidodinium*-3 and 4" sequences in D2 clade, and (iii) "*Lepidodinium*-5, 6 and 7" sequences in D3 clade. Thus, the six sequences described above are concluded as VI-type. The "*Lepidodinium*-1" sequence and sequences of diatoms (and *K. veneficum*) formed a clade with a MLBP of 100% and a BPP of 1.0 (highlighted by an arrowhead in Fig. 3B), suggesting that this version was acquired from a diatom (i.e., LA-type).

Three versions of CPOX were identified in *L. chlorophorum*, but none of them was of VI-type or EA-type. The CPOX phylogeny (Fig. 3B) placed the "*Lepidodinium*-1" sequence in a remote position from the sequences of peridinin-containing dinoflagellates or green algae/land plants. Instead, the *Lepidodinium*-1 sequence grouped with the

bacterial sequences, as well as the eukaryotic sequences for the cytosolic pathway, with a MLBP of 77% and a BPP of 0.87 (highlighted by an arrowhead in Fig. 3C). This protein most likely bears no N-terminal extension (Table S5). Altogether, we propose that the *Lepidodinium*-1 sequence encodes a cytosolic CPOX enzyme involved in C4 pathway, and omitted from the discussion below. The "*Lepidodinium*-2" and "*Lepidodinium*-3" sequences share high sequence similarity in the mature protein region, while their N-terminal regions are distinct from each other (Table S5). In the CPOX phylogeny, the two *Lepidodinium* sequences showed a specific affinity to the diatom sequences (MLBP of 86% and BPP of 0.98; highlighted by a double-arrowhead in Fig. 3C), instead of the dinoflagellate or green plant sequences. Thus, the ancestral CPOX of the two *L. chlorophorum* sequences was most likely of LA-type.

We conclude that *L. chlorophorum* possesses two distinct VI-type (*Lepidodinium*-1 and 3) and a single LA-type PPOX (*Lepidodinium*-2). The PPOX sequences of peridinin-containing dinoflagellates were split into two distinct clades (D1 and D2), and the two clades received strong statistical support from both ML bootstrap and Bayesian analyses (Fig. 4A). The *Lepidodinium*-1 and 3 sequences were included in D1 and D2 clades, respectively. As the PPOX sequences (including the *Lepidodinium*-3 sequence) in D2 clade can be considered as the cytosolic version, we did not discuss the *Lepidodinium*-3 sequence further. The PPOX phylogeny recovered a robust clade comprising the *Lepidodinium*-2 sequence and, the sequences of haptophytes, stramenopiles, *Karlodinium veneficum* and *Karenia brevis* (MLBP of 100% and BPP of 0.99; highlighted by a double-arrowhead in Fig. 4A). Thus, the *Lepidodinium*-2 sequence was acquired from an organism distantly related to dinoflagellates or green algae/land plants (i.e., LA-type).

In *L. chlorophorum*, we identified four versions of FeCH. The "*Lepidodinium*-1" sequence is of VI-type, as this sequence apparently shared the origin with the sequences of peridinin-containing dinoflagellates, and their monophyly was supported with a MLBP of 100% and a BPP of 1.0 (Fig. 4B). On the other hand, we consider the rest of the sequences of *L. chlorophorum* as LA-type, as the "*Lepidodinium*-2," "*Lepidodinium*-3" and "*Lepidodinium*-4" sequences appeared to be distantly related to the dinoflagellate clade described above or the green algae/land plant sequences in the FeCH phylogeny (Fig. 4B). The precise positions of the *Lepidodinium*-2 and *Lepidodinium*-4 sequences were unresolved, and it remains unclear how *L. chlorophorum* acquired the two versions of FeCH. On the other hand, the FeCH phylogeny united the "*Lepidodinium*-3" and *Karenia*-1 sequences together (MLBP of 100% and BPP of 1.0), and this dinoflagellate clade was then connected to two γ-proteobacterial sequences with a MLBP of 75% and a BPP of 0.99 (highlighted by an arrowhead in Fig. 4B). We have already proposed the two scenarios for the origin of the *Lepidodinium*-3 and *Karenia*-1 sequences, in which two lateral gene transfers were invoked (see the previous section for the details).

The most prominent feature in the *L. chlorophorum* pathway is that no gene from the green algal endosymbiont was detected in the heme biosynthesis. Instead, genes transferred from organisms related to neither host (dinoflagellate) nor endosymbiont (green alga) largely contributed to the *L. chlorophorum* pathway. The impact of lateral gene

transfer on the heme biosynthesis is most prominent in the steps catalyzed by PBGD, UROS and CPOX, in which only LA-type proteins were identified.

## Chl *a* biosynthetic pathway

We surveyed the transcripts encoding enzymes involved in the Chl *a* biosynthesis in *Karenia brevis*, *Karlodinium veneficum* and *L. chlorophorum*, and assessed their origins individually. Overall, all the enzymes required to synthesize Chl *a*, except MgPME cyclase, was retrieved from the transcriptomic data from the three dinoflagellates. To our knowledge, no sign for nucleus-encoded MgPME cyclase, which converts MgPME to divinyl protochlorophyllide, has been detected in complex algae (*Wilhelm et al., 2006*; *Nymark et al., 2009*). Although not described in detail, we additionally surveyed the single-subunit MgPME cyclase encoded by *chlE*, which are phylogenetically distinct from the multi-subunit MgPME cyclase, but yielded no significant match. We suspect an as-yet-unidentified enzyme forming E-ring in the photosynthetic eukaryotes described above. Twenty-two out of the 27 transcripts investigated here were found to possess N-terminal extensions, which likely work as plastid-localizing signals (note that 11 sequences were proceeded by the putative SPs, and two of their non-SP portion were predicted to have the TP potential; see Table S5 for the details).

### *Large impact of EGT on the* Karenia brevis *and* Karlodinium veneficum *pathways*

Mg chelatase (MgCH), which comprises three subunits ChlD, ChlH and ChlI, inserts $Mg^{2+}$ to protoporphyrin IX. In *Karlodinium veneficum*, ChlI is plastid-encoded (*Gabrielsen et al., 2011*) and the rest of the subunits were nucleus-encoded (see below). Although no plastid genome data is available for *Karenia brevis*, *chlI* is predicted to reside in the plastid genome of *Karenia mikimotoi* based on a pioneering study on the plastid transcriptome of this species (*Dorrell, Hinksman & Howe, 2016*). In this study, we identified the partial *chlI* contig in the transcriptomic data of *K. brevis* (contig No. 0173787962). The 3′ terminus of this contig was not completed, we could not conclude whether the *chlI* transcript received the poly(U) tail, which is the unique RNA modification occurred in peridinin-containing plastids as well as the non-canonical plastids of *Karenia mikimoti* and *Karlodinium veneficum* (*Dorrell & Howe, 2012*; *Richardson, Dorrell & Howe, 2014*). Although the precise genome harboring *chlI* in *K. brevis* remains uncertain, we assume that the *Karenia brevis chlI* sequence was transcribed from the plastid genome as demonstrated in *Karenia mikimotoi* by *Richardson, Dorrell & Howe (2014)*. As the current study focuses on nucleus-encoded proteins involved in plastid metabolisms, we stopped examining the origin and evolution of ChlI in the two kareniacean species (and *L. chlorophorum*; see below) any further.

We here examine the evolutionary origins of two nucleus-encoded subunits of MgCH, ChlH and ChlD, in *Karenia brevis* and *Karlodinium veneficum*. A single version of ChlD was identified in *Karenia brevis* and *Karlodinium veneficum*. The ChlD phylogeny (Fig. 5A) recovered a clade of the sequences of haptophytes, *Karenia brevis* and

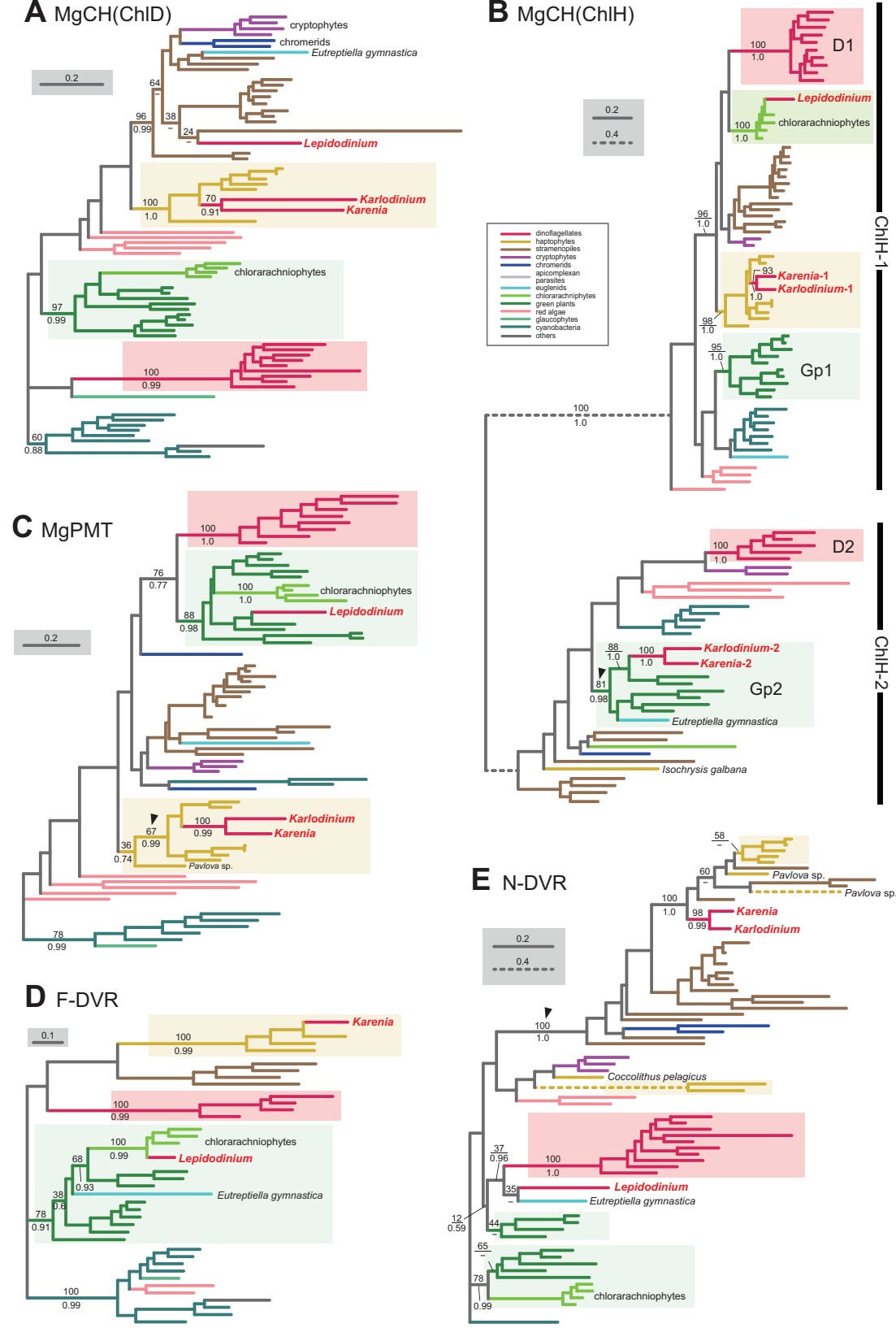

**Figure 5  Maximum-likelihood phylogenies of 5 proteins involved in the Chl *a* biosynthetic pathway.** The details of this figure are same as those of Fig. 2. (A) ChlD, one of the two nucleus-encoded subunits of Mg-chelatase (MgCH). (B) ChlH, the other nucleus-encoded subunit of MgCH. (C) *S*-adenosylmethionine:Mg-protoporphyrin *O*-methyltransferase (MgPMT). (D) divinyl chlorophyllide *a* 8-vinyl-reductase using ferredoxin for electron donor (F-DVR). (E) divinyl chlorophyllide *a* 8-vinyl-reductase using NADPH for electron donor (N-DVR). The identical ML trees with full sequence names and MLBPs ≥ 50% are provided as the supplementary materials. Note that we present no phylogeny of ChlI or MgPME cyclase, as the former is plastid-encoded, and the latter was not identified in dinoflagellates (regardless of plastid-type), diatoms, cryptophytes or haptophytes. The substitution rates of the dashed branches are different from that of the solid lines as indicated in each figure.

*Karlodinium veneficum* with full statistical support, suggesting that the kareniacean ChlD sequences are of EA-type. Both *Karenia brevis* and *Karlodinium veneficum* possess two distinct versions of ChlH (*Karenia*-1 and 2, and *Karlodinium*-1 and 2). ChlH sequences can be split into two distinct clades, "ChlH-1" and "ChlH-2" (Fig. 5B) as a previous study reported (*Lohr, Im & Grossman, 2005*). ChlH-1 sequences are ubiquitously distributed in photosynthetic organisms, while ChlH-2 sequences have been found in restricted lineages. The sequences of green algae/land plants formed two distinct clades (Gp1 and 2 clades). In the ChlH phylogeny (Fig. 5B), the haptophyte ChlH-1 sequences, the *Karenia*-1 and *Karlodinium*-1 sequences formed a clade supported with a MLBP of 98% and a BPP of 1.0. In contrast, the *Karenia*-2 and *Karlodinium*-2 sequences were nested within the Gp2 clade containing ChlH-2 sequences of green algae and a euglenid, and their monophyly received a MLBP of 81% and a BPP of 0.98 (highlighted by an arrowhead in Fig. 5B). Thus, we conclude that *Karenia brevis* and *Karlodinium veneficum* possess ChlH-1 sequences acquired from the haptophyte endosymbiont (i.e., EA-type), while their ChlH-2 sequences are of green algal origin (i.e., LA-type).

*S*-adenosylmethionine: Mg-protoporphyrin *O*-methyltransferase (MgPMT) converts Mg-protoporphyrin IX to Mg protoporphyrin IX monomethyl ester. The MgPMT sequences of *Karenia brevis* and *Karlodinium veneficum* grouped with the haptophyte sequences (except the one of *Pavlova*), and their monophyly was supported by a MLBP of 67% and a BPP of 0.99 (highlighted by an arrowhead in Fig. 5C). The two kareniacean species most likely use the MgPMT acquired from the haptophyte endosymbiont (i.e., EA-type).

No MgPME cyclase has been identified in any dinoflagellates regardless of plastid-type, and we could not examine the origin and evolution of this enzyme (see above). However, DVR, which recognizes the product of MgPME cyclase (divinyl protochlorophyllide) as the substrate and generate protochlorophyllide, were identified in both dinoflagellates bearing peridinin and those bearing non-canonical plastids. We identified both N-DVR and F-DVR sequences in *Karenia brevis*, while only N-DVR sequence was found in *Karlodinium veneficum*. The F-DVR phylogeny (Fig. 5D) recovered the clade of the sequences of *K. brevis* and haptophytes with robust statistical support, suggesting that *K. brevis* was acquired from the haptophyte endosymbiont (i.e., EA-type).

The N-DVR phylogeny (Fig. 5E) united the sequences of *Karenia brevis* and *Karlodinium veneficum* together with a MLBP of 98% and a BPP of 0.99, and the kareniacean clade showed no clear phylogenetic affinity to either haptophyte sequences or other dinoflagellate sequences. Instead, the kareniacean clade grouped with the sequences

of stramenopiles, haptophytes, and two chromerids (*Chromera* and *Vitrella*) supported with a MLBP of 100% and a BPP of 1.0 (highlighted by an arrowhead in Fig. 5E). In this large clade, the affinity between the kareniacean clade and haptophyte sequences remained uncertain, and we are not sure of the precise origin of the two kareniacean sequences.

In land plants, conversion of protochlorophyllide to chlorophyllide *a* is catalyzed by the light-dependent and/or light-independent forms of POR. The light-dependent POR is nucleus-encoded, while the light-independent form comprises three plastid-encoded subunits (ChlB, ChlL and ChlN). No gene for light-independent POR was found in the sequenced region of the *Karlodinium veneficum* plastid genome (Gabrielsen et al., 2011) or plastid transcriptomic data of *Karenia mikimotoi* (Dorrell, Hinksman & Howe, 2016), implying that kareniacean species lack the light-independent version. We identified two and three distinct versions of the light-dependent POR in *Karenia brevis* and *Karlodinium veneficum*, respectively (*Karenia*-1 and 2, and *Karlodinium*-1-3), as demonstrated in Hunsperger, Randhawa & Cattolico (2015). In the POR phylogeny (Fig. 6A), the *Karenia*-1 and *Karlodinium*-1 sequences grouped with the sequences of stramenopiles, cryptophytes and haptophytes, and their monophyly was supported by a MLBP of 95% and a BPP of 0.98 (highlighted by an arrowhead in Fig. 6A). The *Karenia*-1 and *Karlodinium*-1 sequences cannot be of VI-type, as the two sequences are distantly related to other dinoflagellate sequences included in the alignment. However, it is difficult to classify the *Karenia*-1 and *Karlodinium*-1 sequences into EA-type or LA-type, as the relationship between the two dinoflagellate sequences and the haptophyte sequences was unresolved in the particular clade. Thus, we leave the origins of the two sequences uncertain in this study. The *Karenia*-2, *Karlodinium*-2 and *Karlodinium*-3 sequences robustly grouped together within the haptophyte sequences, and this "haptophyte" clade received a MLBP of 98% and a BPP of 0.99 (Fig. 6A). Thus, we conclude that these sequences were acquired from the haptophyte endosymbiont (i.e., EA-type).

The final step of the Chl *a* biosynthesis is catalyzed by Chl synthase (CS). A single version of CS was identified in *Karenia brevis* and *Karlodinium veneficum*. The two kareniacean sequences were placed within the haptophyte clade in the CS phylogeny, and the "haptophyte" clade as a whole was supported by a MLBP of 98% and a BPP of 1.0 (Fig. 6B). Thus, the sequences of *Karenia brevis* and *Karlodinium veneficum* are considered as EA-type.

The phylogenetic analyses described above revealed that EA-type proteins operate in all the steps converting protoporphyrin IX to Chl *a* in *Karlodinium veneficum* and/or *Karenia brevis* (except the step catalyzed by MgPME cyclase; see above). In addition, the common ancestor of *Karenia brevis* and *Karlodinium veneficum* should have possessed LA-type ChlH-2, POR and N-DVR, which were acquired from phylogenetically diverse eukaryotes distantly related to dinoflagellates or haptophytes.

### Genetic influx from phylogenetically diverse organisms shaped the *L. chlorophorum* pathway

As discussed in the previous section, the Chl *a* biosynthetic pathway in *Karenia brevis* and *Karlodinium veneficum* appeared to be shaped by the genes transferred from the

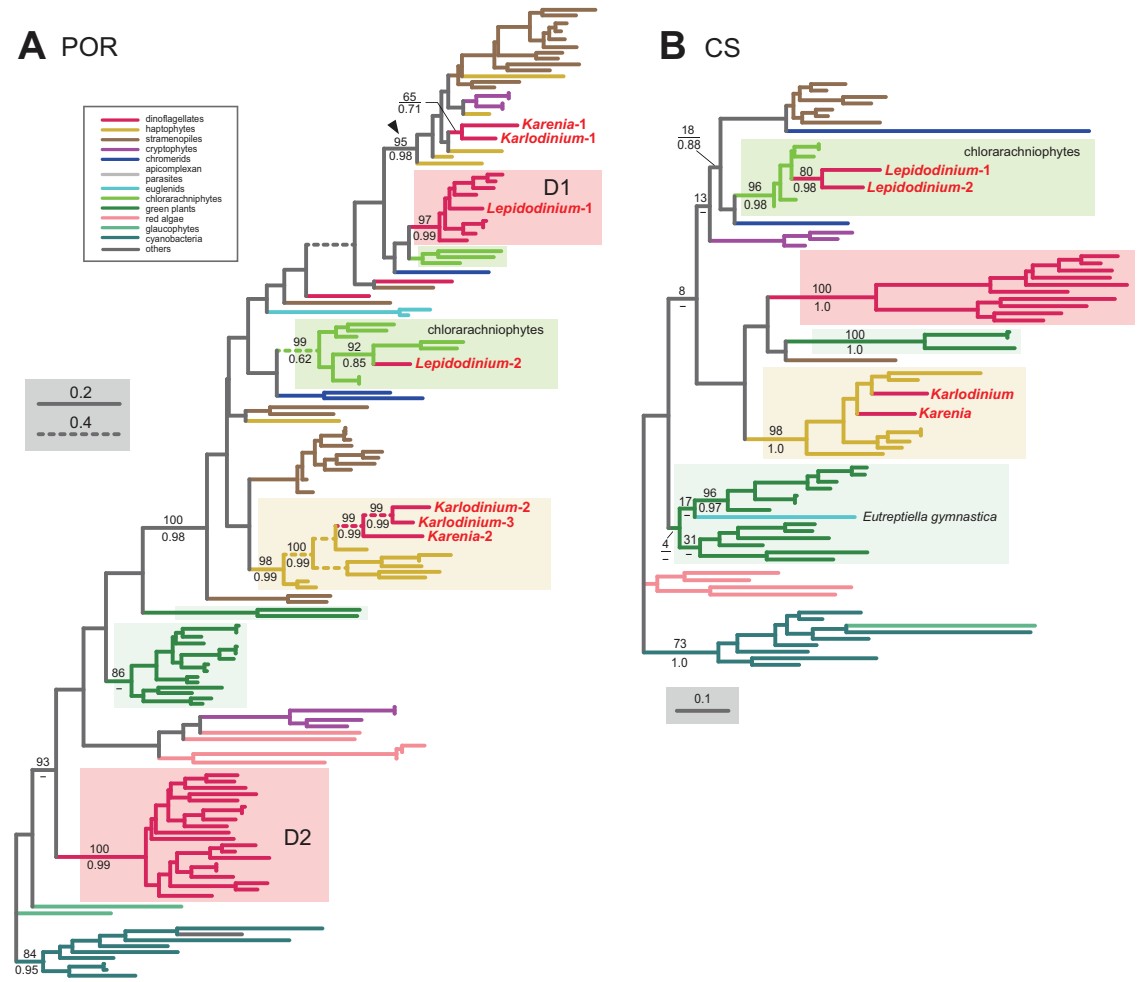

**Figure 6 Maximum-likelihood phylogenies of two proteins involved in the Chl *a* biosynthetic pathway.** The details of this figure are same as those of Fig. 2. (A) light-dependent protochlorophyllide reductase (POR). (B) chlorophyll synthase (CS).

endosymbiont (i.e., a haptophyte in the above systems). Curiously, this is not the case for the same pathway in *L. chlorophorum*, of which plastid was derived from a pedinophyte green alga. Note that we present no result from the *L. chlorophorum* ChlI, which turned out to be plastid-encoded (*Kamikawa et al., 2015a*). We identified eight proteins involved in the Chl *a* biosynthetic pathway in *L. chlorophorum*, and assess their phylogenetic origins individually (Figs. 5A–5E and 6A–6B). Among the eight proteins examined here, we conclude MgPMT as a sole EA-type protein among those involved in the *L. chlorophorum* pathway. The MgPMT phylogeny (Fig. 5C) placed the *L. chlorophorum* sequence within a radiation of the sequences of green algae, land plants and chlorarachniophytes, and their monophyly was supported by a MLBP of 88% and a BPP of 0.98.

Our surveys and phylogenetic analyses revealed that VI-type proteins were almost entirely eliminated from the *L. chlorophorum* pathway. We identified only one of the two versions of POR (*Lepidodinium*-1) as VI-type. The POR phylogeny (Fig. 6A)

recovered two distinct clades of the sequences of peridinin-containing dinoflagellates (D1 and D2 clades), and placed the *Lepidodinium*-1 sequence within D1 clade. D1 clade containing the *Lepidodinium*-1 sequence as a whole received a MLBP of 97% and a BPP of 0.99.

We could not clarify the origin of the N-DVR sequence of *L. chlorophorum*, of which position was resolved in neither ML nor Bayesian phylogenetic analyses (Fig. 5E). The *Lepidodinium* sequence was excluded from the robust clade of the sequences of peridinin-containing dinoflagellates, suggesting that this sequence cannot be of VI-type. However, it is difficult to pursue the origin of the *Lepidodinium* sequence any further, as the N-DVR phylogeny failed to exclude a potential affinity between the sequence of interest and the green algal/land plant sequences (Fig. 5E).

We classified ChlD, ChlH, one of the two versions of POR (*Lepidodinium*-2), F-DVR and CS into LA-type. In the ChlD phylogeny, the sequence of *L. chlorophorum* appeared to be excluded from the sequences of peridinin-containing dinoflagellates and those of the green algal/land plant sequences (Fig. 5A), suggesting that this sequence cannot be of VI-type or EA-type. Instead, the sequence of *L. chlorophorum*, as well as those of chromerids, cryptophytes and *Eutreptiella*, were placed within the radiation of the stramenopile sequences, and their monophyly was supported by a MLBP of 96% and a BPP of 0.99. This tree topology prompts us to propose that *L. chlorophorum* ChlD was acquired from a stramenopile (i.e., LA-type).

To our surprise, the ChlH, POR and CS phylogenies placed the sequences of *L. chlorophorum* within the radiation of the chlorarachniophyte sequences, and their monophylies received MLBPs > 96% and by BPPs > 0.62 (Figs. 5B, 6A and 6B). We here propose multiple gene transfers from chlorarachniophytes to the ancestral *Lepidodinium* cell to interpret the aforementioned tree topologies. The putative chlorarachniophyte origins of the three genes are not contradict to a recent molecular clock analysis, in which the green algal endosymbiosis leading to the *Lepidodinium* plastids was predicted to occur more recently than that leading to the chlorarachniophyte plastids (*Jackson et al., 2018*).

The F-DVR phylogeny reconstructed a robust affinity between the sequence of *L. chlorophorum* and the chlorarachniophyte clade within the land plant/green algal sequences (Fig. 5D). This tree topology requires a combination of an EGT and LGT, as *Lepidodinium* species and chlorarachniophytes acquired their plastids commonly from green algae. Either ancestral *Lepidodinium* cell or the ancestral chlorarachniophyte acquired an F-DVR gene from the green algal endosymbiont (i.e., EGT), followed by the second gene transfer between the two organisms (i.e., LGT). Thus, we have to leave the evolutionary origin of the *L. chlorophorum* F-DVR uncertain.

The phylogenetic analyses described above revealed that EGT was much less significant in the *L. chlorophorum* pathway than the kareniacean pathway (see above). Instead, chlorarachniophytes seemingly donated the genes encoding the proteins involved in three out of the five steps in the *L. chlorophorum* pathway.

## Non-mevalonate pathway for the IPP biosynthesis

The origin of all the enzymes involved in the non-mevalonate pathway of *Karenia brevis* and *Karlodinium veneficum* were investigated carefully in *Bentlage et al. (2016)*. On the other hand, the entire picture of the *L. chlorophorum* pathway remain to be completed, leaving five out the seven enzymes involved in this pathway unidentified (*Minge et al., 2010*). We successfully identified all the enzymes involved in the non-mevalonate pathway in *L. chlorophorum* (see below). In this section, we mainly examined the origins of individual enzymes involved in the non-mevalonate pathway of *L. chlorophorum*, coupled with a brief overview of the same pathway of the two kareniacean species. Thirty-one out of the 32 transcripts investigated here were found to possess N-terminal extensions, which likely work as plastid-localizing signals (note that 12 sequences were proceeded by the putative SPs, and three of the non-SP portions were predicted to have the TP potential; see Table S5 for the details).

For the step synthesizing 1-deoxy-D-xylulose-5-phosphate (DXP) from pyruvate and glyceraldehyde 3-phosphate, *L. chlorophorum* was found to possess two versions of DXP synthase (DXS) (*Lepidodinium*-1 and 2). The DXS phylogeny robustly grouped the *Lepidodinium*-1 and 2 sequences with those of peridinin-containing dinoflagellates and *K. brevis* (Fig. 7A). The sequence of *K. veneficum* was found to be remote from the dinoflagellate clade, and showed an affinity to the haptophyte sequences (Fig. 7A). The clade comprising the sequences of *K. veneficum* and haptophytes received a MLBP of 63% and a BPP of 0.82 (if the *Pavlova* sequence was excluded, the "*Karlodinium* + haptophyte" clade was supported by a MLBP of 98% and a BPP of 1.0). Thus, we conclude that the DXS sequences of *L. chlorophorum* and *Karenia brevis* were of VI-type, while that of *Karlodinium veneficum* was of EA-type.

The conversion of DXP to 2-C-methyl-D-erythritol 4-phosphate (MEP) is catalyzed by DXP reductase (DXR). *Minge et al. (2010)* detected a partial sequence of a VI-type DXR (GenBank accession number CCC15090). From our transcriptome data, two versions of DXR were identified in *L. chlorophorum* (*Lepidodinium*-1 and 2; the former corresponds to the previously reported DXR sequence). The DXR phylogeny (Fig. 7B) reconstructed a robust monophyly of the *Lepidodinium*-1 and 2 sequences, the sequences of peridinin-containing dinoflagellates, and those of *Karenia brevis* and *Karlodinium veneficum* (MLBP of 100% and BPP of 1.0), suggesting that *L. chlorophorum* and the two kareniacean species use VI-type proteins for this reaction.

*L. chlorophorum* was found to possess both LA-type and VI-type versions of MEP cytidylyltransferase (IspD) (*Lepididodinium*-1 and 2) to convert MEP into 4-(cytidine 5′-diphospho)-2-C-methyl-D-erythritol (CDP-ME). The *Lepidodinium*-1 sequence was concluded as LA-type based on its remote position from the sequences of peridinin-containing dinoflagellates or green plants in the IspD phylogeny (Fig. 7C). Instead, the *Lepidodinium*-1 sequence showed a specific affinity to the sequence of a stramenopile *Ochromonas* sp. with a MLBP of 100% and a BPP of 1.0. As the monophyly of the stramenopile sequences was not recovered probably due to insufficient phylogenetic signal in the IspD alignment, two scenarios remain possible. The simplest scenario

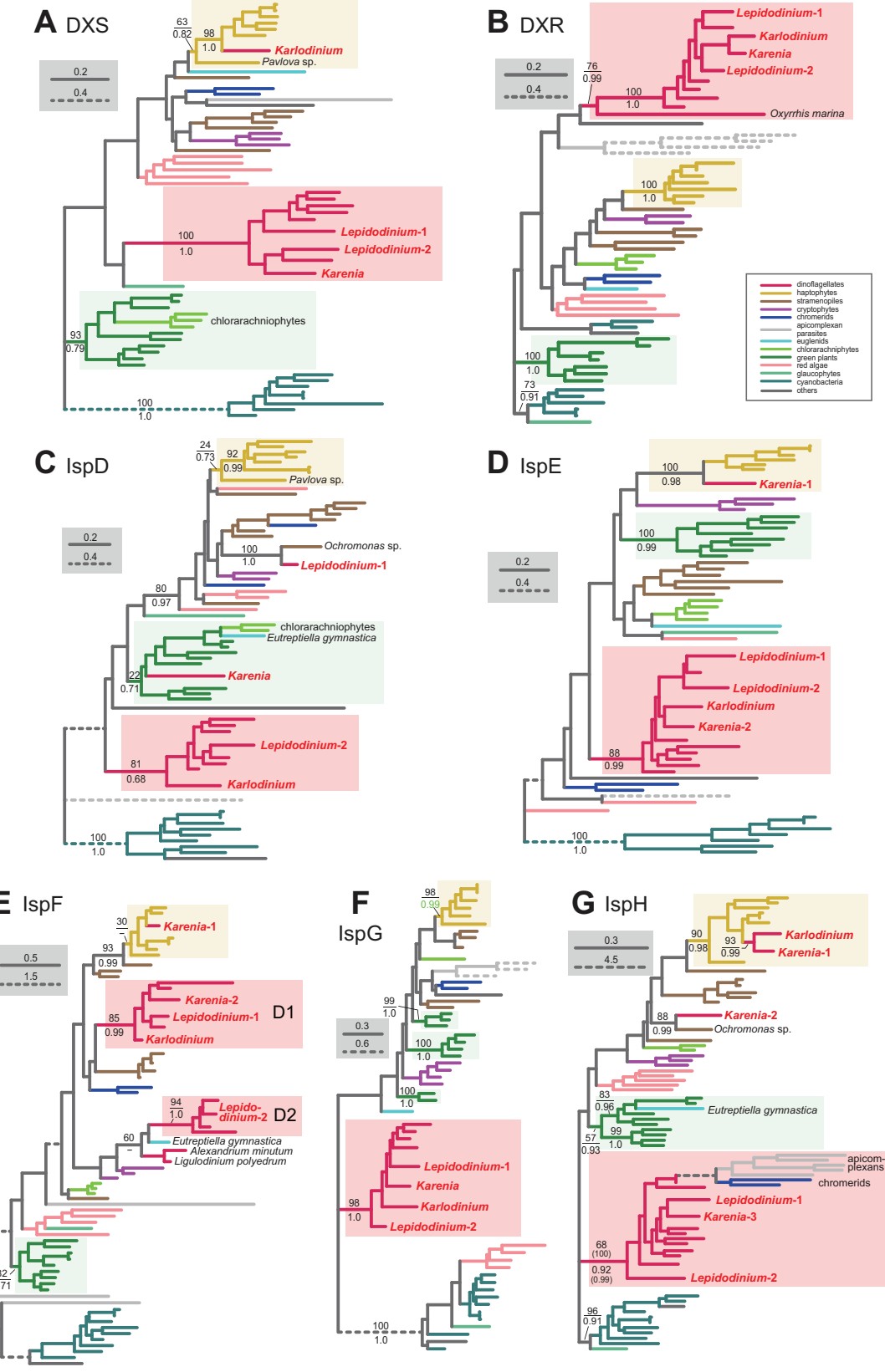

**Figure 7 Maximum-likelihood phylogenies of seven proteins involved in the non-mevalonate pathway for IPP biosynthesis.** The details of this figure are same as those of Fig. 2. (A) 1-deoxy-D-xylulose-5-phosphate (DXP) synthase (DXS). (B) DXP reductoisomerase (DXR). (C) 2-C-methyl-D-erythritol 4-phosphate cytidylyltransferase (IspD). (D) 4-diphosphocytidyl-2-C-methyl-D-erythritol kinase (IspE). (E) 2-C-methyl-D-erythritol 2,4-cyclodiphosphate synthase (IspF). (F) 1-hydroxy-2-methyl-2-butenyl 4-diphosphate (HMB-PP) synthase (IspG). (G) HMB-PP reductase (IspH). The identical ML trees with full sequence names and MLBPs ≥ 50% are provided as the Supplementary Materials. Note that the substitution rates of the dashed branches are different from that of the solid lines as indicated in each figure.

assumes that the *ispD* gene was transferred between *L. chlorophorum* and *Ochromonas* sp. Alternatively, a scenario, in which invokes separate gene transfers from an as-yet-unknown organism to *L. chlorophorum* and *Ochromonas* sp., is also possible. To examine the two scenarios in the future, we need to prepare and phylogenetically analyze a new IspD alignment including the sequences from stramenopiles closely related to *Ochromonas* sp. (i.e., chrysophycean algae) in the future. In contrast, the *Lepidodinium*-2 sequence, together with a single sequence of *K. veneficum*, were considered as VI-type, as they formed a clade with those of peridinin-containing dinoflagellates (MLBP of 81% and BPP of 0.68). The IspD sequence of *Karenia brevis* appeared to be nested within the clade of green algae/land plants and chlorarachniophytes, and being distantly related to the dinoflagellate clade (including the sequences of *L. chlorophorum* and *Karlodinium veneficum*) or the haptophyte clade (Fig. 7C). The position of *Karenia brevis* IspD is consistent with the green algal origin of this enzyme proposed by *Bentlage et al. (2016)*.

*Minge et al. (2010)* reported a VI-type 4-diphosphocytidyl-2-C-methyl-D-erythritol kinase (IspE) (GenBank accession number CCC15094), which phosphorylates CDP-ME to 2-phospho-4-(cytidine 5′-diphospho)-2-C-methyl-D-erythritol (CDP-MEP), in *L. chlorophorum*. In this study, we detected two distinct versions of IspE (*Lepidodinium*-1 and 2)—the *Lepidodinium*-1 sequence corresponds to the version reported in *Minge et al. (2010)* and the *Lepidodinium*-2 sequence is a novel version of IspE. The IspE phylogeny (Fig. 7D) recovered a monophyletic clade of the two sequences of *L. chlorophorum*, the sequences of peridinin-containing dinoflagellates, the sequence of *Karlodinium veneficum*, and one of the two sequences of *Karenia brevis* (*Karenia*-2), which was supported by a MLBP of 88% and a BPP of 0.99. Thus, *L. chlorophorum*, *Karenia brevis* and *Karlodinium veneficum* possess VI-type versions of IspE. In addition, *K. brevis* possesses an EA-type IspE (*Karenia*-1), which was united with the haptophyte sequences with a MLBP of 100% and a BPP of 0.98.

The ancestral dinoflagellate likely possessed two versions of 2-C-methyl-D-erythritol 2,4-cyclodiphosphate synthase (IspF), which converts CDP-MEP to 2-C-methyl-D-erythritol 2,4-cyclodiphosphate (MEcPP). In the IspF phylogeny (Fig. 7E), the vast majority of the dinoflagellates sequences was split into two clades (D1 and D2), of which monophylies were supported by MLBPs of 85–94% and BPPs of 0.99–1.0, respectively. D1 clade appeared to contain one of the two sequences of *L. chlorophorum* (*Lepidodinium*-1), as well as the sequence of *Karlodinium veneficum* and one of the two sequences of *Karenia brevis* (*Karenia*-2). The other version of *L. chlorophorum* (*Lepidodinium*-2) was nested within D2 clade. Thus, we conclude that *L. chlorophorum*, *Karenia brevis* and *Karlodinium veneficum* possess VI-type versions of IspF. As reported in *Bentlage et al. (2016)*, *K. brevis*

possesses an additional IspF sequence (*Karenia*-1) with a phylogenetic affinity to the haptophyte sequences, suggesting that this version is of EA-type.

The origin and evolution of 1-hydroxy-2-methyl-2-butenyl 4-diphosphate (HMB-PP) synthase (IspG), which synthesize HMB-PP from MEcPP, seems straightforward in dinoflagellates. We phylogenetically analyzed two versions of IspG in *L. chlorophorum* (*Lepidodinium*-1 and 2) identified in this study, together with the sequences of *Karenia brevis* and *Karlodinium veneficum*. In the IspG phylogeny (Fig. 7F), the aforementioned dinoflagellate sequences tightly clustered with the sequences of peridinin-containing dinoflagellates (MLBP of 98% and BPP of 1.0). Thus, we concluded that *Karenia brevis*, *Karlodinium veneficum* and *L. chlorophorum* uses VI-type enzymes to synthesize MEcPP.

The last step of the non-mevalonate pathway is catalyzed by HMB-PP reductase (IspH) to yield IPP from HMG-PP. We identified two versions of IspH in *L. chlorophorum* (*Lepidodinium*-1 and 2). The IspH phylogeny (Fig. 7G) recovered a clade comprising the sequences of peridinin-containing dinoflagellates, apicomplexan parasites and chromerids, as well as the two versions of *L. chlorophorum* and one of the three versions of *K. brevis* (*Karenia*-3), with a MLBP of 68% and a BPP of 0.92. We noticed the rapidly evolving natures of the apicomplexan and chromerid sequences, which highly likely biased the phylogenetic inference from the IspH alignment. Consequently, the same alignment was re-analyzed after the exclusion of the long-branch sequences. In both ML and Bayesian analyses of the second IspH alignment, the *Lepidodinium*-1 and 2, and *Karenia*-3 sequences grouped with those of peridinin-containing dinoflagellates with a MLBP of 100% and a BPP of 0.99 (Fig. 7G). We conclude that the two versions of IspH in *L. chlorophorum* are of VI-type. *K. brevis* was found to possess two additional versions of IspH (*Karenia*-1 and *Karenia*-2). The IspH phylogeny (Fig. 7G) united the *Karenia*-1 sequence and a single IspH sequence of *Karlodinium veneficum* with the sequences of haptophytes with a MLBP of 90% and a BPP of 0.98. The *Karenia*-2 sequence was excluded from the dinoflagellate or haptophyte clade, and instead connected to the sequence of a stramenopile *Ochromonas* sp. with a MLBP of 88% and a BPP of 0.99 (Fig. 7G). As the sequence of *Ochromonas* sp. showed no clear affinity to other stramenopile sequences, the origin of the *Karenia*-2 sequence remains unclear. Thus, as discussed in *Bentlage et al. (2016)*, *Karenia brevis* uses VI-type (*Karenia*-3), EA-type (*Karenia*-1) and LA-type (*Karenia*-2) enzymes to yield IPP, while *Karlodinium veneficum* possesses a single, EA-type version.

As *Bentlage et al. (2016)* demonstrated, the non-mevalonate pathways in *Karenia brevis* and *Karlodinium veneficum* are evolutionary hybrids of VI-type and EA-type enzymes. In sharp contrast, the same pathway in *L. chlorophorum* appeared to be dominated by VI-type proteins, except a single LA-type protein (one of the two IspD versions).

# DISCUSSION

## Perspectives toward the evolution of kareniacean dinoflagellates and their plastids

As anticipated from the haptophyte origin of the kareniacean plastids, the *Karenia brevis* and *Karlodinium veneficum* pathways investigated here appeared to be composed of

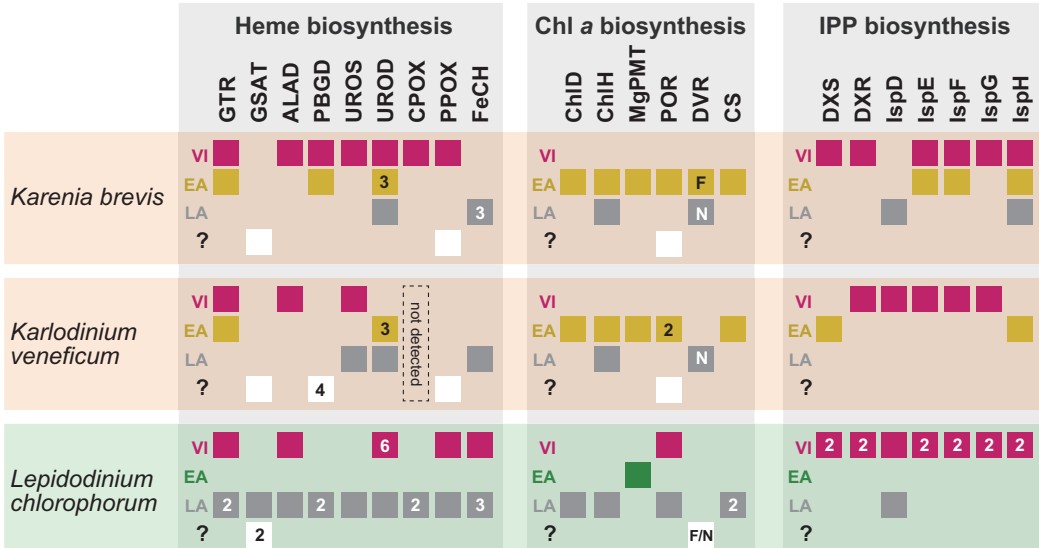

**Figure 8 Overview of the origins of proteins involved in three plastid-localized biosynthetic pathways in *Karenia brevis*, *Karlodinium veneficum* and *L. chlorophorum*.** The origins of proteins of interest were classified into three types: (i) "VI-type" which were vertically inherited from the ancestral dinoflagellate beyond haptophyte/green algal endosymbiosis, (ii) "EA-type" which were acquired from the endosymbiont, and (iii) "LA-type" which were acquired from organisms distantly related to the host (dinoflagellates) or endosymbiont (haptophytes or green algae). Squares indicate the numbers and types of proteins of interest in the three dinoflagellates. In case of multiple versions being identified in one species, the numbers of the versions are shown in the corresponding squares. The squares are color-coded as follows: red, VI-type protein; yellow, EA-type in *Karenia brevis* and *Karlodinium veneficum*; green, EA-type in *Lepidodinium*; grey, LA-type. For DVR involved in the Chl *a* biosynthetic pathway, we distinguish N-DVR and F-DVR by labeling "N" and "F," respectively. The open squares in the fourth lows labeled with question marks represent the sequences of which origins remain uncertain.

three evolutionary types of proteins, namely EA-, VI- and LA-types (*Patron, Waller & Keeling, 2006*; *Nosenko et al., 2006*; *Hunsperger, Randhawa & Cattolico, 2015*; *Bentlage et al., 2016*). Nevertheless, the impact of the genetic influx from the haptophyte endosymbiont was different among the three pathways in *Karenia brevis/Karlodinium veneficum* (Fig. 8). In the two kareniacean species, EA-type proteins, together with a few LA-type proteins, found to dominate the Chl *a* biosynthesis, albeit no VI-type protein was detected. In sharp contrast, VI-type proteins persist in five to six out of the seven steps required for the non-mevalonate pathway for the IPP biosynthesis, albeit the contributions of EA-type and LA-type proteins may not be negligible. The evolutionary chimerism is most advanced in heme biosynthesis, in which all the three protein types are identified (Fig. 8).

Based on the difference in degree of evolutionary chimerism among the three pathways discussed in the previous section, we here explore the early evolution of kareniacean species. *Saldarriaga et al. (2001)* hinted non-photosynthetic intermediate stages for the dinoflagellates bearing non-canonical plastids based on the dinoflagellate phylogeny inferred from small subunit ribosomal DNA sequences, albeit they made no mention of the presence/absence of a residual plastid. *Patron, Waller & Keeling (2006)* and *Nosenko et al. (2006)* hypothesized that the ancestral kareniacean species possessed a non-photosynthetic plastid prior to the haptophyte endosymbiosis, although both

studies assessed a restricted number of plastid-localized proteins. The above proposal is plausible, as secondarily non-photosynthetic eukaryotes often possess plastids with no photosynthetic activity but diverse metabolic capacities including those to synthesize heme and/or IPP (*Lim & McFadden, 2010*; *Lohr, Schwender & Polle, 2012*; *Kamikawa et al., 2015b*, *2015c*, *2017*; *Janouškovec et al., 2017*). Notably, Kareniaceae includes a kleptoplastic species found in the Ross Sea, Antarctica, in addition to the species bearing haptophyte-derived plastids (*Gast et al., 2006*, *2007*). The coexistence of the species bearing the haptophyte-derived non-canonical plastids and that leading a kleptoplastic lifestyle in the same family lends an additional support for the non-photosynthetic nature of their common ancestor. The hypothesis for the non-photosynthetic nature in the ancestral kareniacean species can explain well the elimination of VI-type proteins from the Chl *a* biosynthesis in both *Karenia brevis* and *Karlodinium veneficum* (Fig. 8). During the putative non-photosynthetic period in the early kareniacean evolution, the proteins involved in the Chl *a* biosynthesis may have been dispensable, leading to discard of the corresponding genes from the dinoflagellate genome. Interestingly, *Patron, Waller & Keeling (2006)* identified *atpC* transcript encoding a subunit of VI-type ATP synthase in *K. veneficum*, implying that ATP synthase on plastid thylakoid membrane has not been discarded during tertiary endosymbiosis. Nevertheless, the presence of VI-type ATP synthase in the current *K. veneficum* plastid is not necessary to negate the putative non-photosynthetic period for the early kareniacean evolution, as the plastid ATP synthase can persist in secondarily non-photosynthetic organisms (*Donaher et al., 2009*; *Wicke et al., 2013*; *Kamikawa et al., 2015c*, *2017*). An equivalent observation was also reported in a non-photosynthetic cyanobacterium by *Nakayama et al. (2014)*. In the later kareniacean evolution, the entire pathway for the Chl *a* biosynthesis was most likely reconstructed in the haptophyte-derived plastid by incorporating exogenous genes (acquired mainly from the endosymbiont). In contrast, both *Karenia brevis* and *Karlodinium veneficum* seemingly use VI-type proteins to synthesize both heme and IPP, suggesting that the proteins originally worked in the peridinin-containing plastid persisted in the ancestral kareniacean species beyond the haptophyte endosymbiosis. The two pathways have been modified after the haptophyte endosymbiosis by incorporating exogenous genes acquired from phylogenetically diverse organisms (including the endosymbiont), as we observed both EA- and LA-type proteins in the current pathways in both *Karenia brevis* and *Karlodinium veneficum* (Fig. 8).

We propose that the putative non-photosynthetic plastid in the ancestral kareniacean species possessed the residual genome as argued below. Polyuridylation of the 3′ termini of plastid transcripts was found primarily in peridinin-containing dinoflagellates, but also *Karenia brevis* and *Karlodinium veneficum* (*Dorrell & Howe, 2012*; *Richardson, Dorrell & Howe, 2014*; *Dorrell, Hinksman & Howe, 2016*). Thus, the machinery for the RNA modification have been inherited from the photosynthetic ancestor bearing a peridinin-containing plastid to the extent kareniacean species beyond the putative non-photosynthetic period. In line with the above speculation, the putative non-photosynthetic plastid in the ancestral kareniacean species most likely retained a transcriptionally active genome.

We can retrieve an additional insight into the early kareniacean evolution by comparing the phylogenetic inventories of VI-, EA- and LA-type proteins in the heme, Chl *a* and IPP biosynthetic pathways between *Karenia brevis* and *Karlodinium veneficum*. For instance, the contribution of VI-type proteins to the heme biosynthesis seemingly differs between *Karenia brevis* and *Karlodinium veneficum*, which retain six and three VI-type proteins for the nine steps in the heme biosynthesis, respectively (Fig. 8). Coincidently, the significance of LA-type proteins in the particular pathway seems to be expanded in the *Karlodinium veneficum* pathway comparing to the *Karenia brevis* pathway. These observations suggest that the reconstruction of metabolic pathways in the haptophyte-derived plastids (i.e., gene acquisitions/losses) was not completed before the separation of the genera *Karenia* and *Karlodinium*. However, we need to assess carefully whether the differences between the *Karenia brevis* and *Karlodinium veneficum* pathways observed in our comparisons stemmed from the incomplete coverages of gene repertories in the two kareniacean species in future studies. Alternatively, the differences between the *Karenia brevis* and *Karlodinium veneficum* pathways observed in our comparisons might reflect the separate haptophyte endosymbioses in *Karenia* and *Karlodinium*, as a plastid small subunit ribosomal DNA (pl-SSU rDNA) phylogeny placed the two dinoflagellate species in two remote positions within the haptophyte clade (Gast et al., 2007). The paraphyly of *Karenia* and *Karlodinium* implied by the pl-SSU rDNA phylogeny is apparently contradict to phylogenetic analyses based on multiple plastid-encoded proteins, in which *Karenia brevis* and *Karlodinium veneficum* grouped together by excluding haptophytes (Gabrielsen et al., 2011; Dorrell & Howe, 2012). The precise origin (or origins) of haptophyte-derived plastids needs to examine by plastid genome-based multigene alignments including broader members of Kareniaceae (e.g., *Takayama* spp.) and haptophytes in the future.

## Perspectives toward the evolution of *Lepidodinium* and its plastids

The phylogenetic inventories of VI-, EA- and LA-type proteins appeared to be different among the heme, Chl *a* and IPP biosynthetic pathways in *L. chlorophorum* (Fig. 8). The IPP synthesis in this species retains VI-type proteins in all of the seven steps, and no EA-type protein was found. In sharp contrast, the contribution of LA-type proteins to the Chl *a* biosynthesis is likely much greater than that of VI- or EA-type proteins. The heme biosynthesis appeared to be distinct from the two pathways described above, as we detected both VI- and LA-type proteins but no EA-type protein. Interestingly, the trend, of which VI-type proteins contribute to the heme and IPP biosyntheses at much greater magnitudes than the Chl *a* biosynthesis, is common between *L. chlorophorum* and *Karenia brevis*/*Karlodinium veneficum* (Fig. 8). Thus, as discussed the putative ancestral state of kareniacean species (see above), we speculate that the ancestral *Lepidodinium*, which engulfed a green algal endosymbiont, experienced a non-photosynthetic period and discarded most of the genes encoding proteins involved in the Chl *a* biosynthesis, but retained a non-photosynthetic plastid with the capacities for synthesizing both heme and IPP. Ferredoxin-NADP$^+$ reductase (FNR) involved in photosynthetic electron transport chain has been identified in photosynthetic, as well as

secondarily non-photosynthetic organisms (*Balconi et al., 2009*; *Nakayama et al., 2014*). Thus, the transcript encoding plastid-type FNR identified in the *L. chlorophorum* transcriptome (*Minge et al., 2010*) does not contradict the putative non-photosynthetic period in the ancestral *Lepidodinium* species.

We unexpectedly revealed a potentially large contribution of chlorarachniophyte genes to the Chl *a* biosynthesis in *L. chlorophorum* (Fig. 8). In the organismal tree of eukaryotes, chlorarachniophytes and dinoflagellates belong to two distantly related taxonomic assemblages, Rhizaria and Alveolata, respectively. Likewise, the current plastids in chlorarachniophytes and *Lepidodinium* were derived from distinct green algal groups, ulvophytes and pedinophytes, respectively (*Suzuki et al., 2016*; *Kamikawa et al., 2015a*). Thus, the relationship between their host lineages or that between their endosymbiont lineages (plastids) can provide no ground for the presence of chlorarachniophyte genes in the *L. chlorophorum* genome. If the ancestral *Lepidodinium* cell fed on chlorarachniophytes in the natural environment, such predator-prey relationship led to the genetic influx from the prey (chlorarachniophyte) genome to the predator (*Lepidodinium*) genome. Nevertheless, under the circumstance postulated above, gene transferred from chlorarachniophytes could not have been restricted to a single metabolic pathway. To understand the biological reasons for the genetic contribution from chlorarachniophytes to the Chl *a* biosynthesis in *L. chlorophorum*, we need to explore (i) potential interaction between dinoflagellates and chlorarachniophytes in the natural environment and (ii) the biochemical and/or physiological commonality in the proteins involved in the Chl *a* biosynthesis between chlorarachniophytes and *L. chlorophorum*. Nevertheless, our proposal for the presence of the "chlorarachniophyte genes" in the extant *L. chlorophorum* genome, which incorporates lateral gene transfers, needs to be re-assessed in the future. In particular, much broader sampling of ChlH, POR, CS and F-DVR sequences would be critical to infer the precise origins of the four "LA-type" proteins involved in the *L. chlorophorum* Chl *a* biosynthesis. Depending on the future ChlH, POR, CS and F-DVR phylogenies with improved sequence sampling, we may need to revise their phylogenetic origins proposed in the current study.

We also noticed a clear difference in contribution of EA-type proteins to the three pathways between *L. chlorophorum* and *Karenia brevis/Karlodinium veneficum* (Fig. 8). EA-type proteins are most likely indispensable for the heme, Chl *a* and IPP biosyntheses in *Karenia brevis/Karlodinium veneficum*. On the other hand, only MgPMT in the Chl *a* biosynthetic pathway appeared to be of green algal origin in *L. chlorophorum*. One potential factor, which could introduce the marked difference between the *L. chlorophorum* and *Karenia brevis/Karlodinium veneficum* pathways, is the difference in plastid-targeting signal of nucleus-encoded plastid-related proteins between their endosymbionts. In both eukaryotes bearing primary plastids (e.g., green algae) and those bearing complex plastids (e.g., haptophytes and dinoflagellates), the vast majority of plastid-related proteins are nucleus-encoded, which are synthesized in the cytosol and localized to the plastid. In green algae, nucleus-encoded plastid-related proteins are synthesized with N-terminal extensions (i.e., transit peptides or TP), which act as the "tags" to pass through the two membranes surrounding their plastids

(*Patron & Waller, 2007*). On the other hand, the "tag" sequences, which enable nucleus-encoded proteins to localize in complex plastids surrounded by three or four membranes, are more complicated than green algal plastids (*Bolte et al., 2009*). *Patron et al. (2005)* revealed that dinoflagellates with peridinin-containing plastids and haptophytes appeared to share a bipartite structure of plastid-targeting signal, which is composed of SP and the TP-like region. Consequently, without any substantial modification on plastid-targeting signals, the ancestral kareniacean species could have targeted the proteins encoded by endosymbiotically transferred genes back to the haptophyte-derived plastid. In contrast, nucleus-encoded plastid-related proteins in the green algal endosymbiont engulfed by the ancestral *Lepidodinium* unlikely possessed bipartite plastid-targeting signals. Thus, in the ancestral *Lepidodinium*, the proteins encoded by endosymbiotically transferred genes needed to acquire bipartite plastid-targeting signals to be localized in the green alga-derived plastid surrounded by four membranes. Altogether, we here propose the initial presence/absence of bipartite plastid-targeting signals was one of the major factors affecting the EGT in dinoflagellates bearing non-canonical plastids. As anticipated from the above scenario, many LA-type proteins acquired from diverse eukaryotes bearing complex plastids (e.g., stramenopiles and chlorarachniophytes) were identified in the heme, Chl *a* and IPP biosynthetic pathways in *L. chlorophorum*. Nevertheless, the factor discussed above may not be dominant enough to exclude EA-type proteins from a green alga (e.g., MgPMT; Fig. 5C) and LA-type proteins from bacteria (e.g., GSAT; Fig. 2B) from the plastid proteome in *L. chlorophorum*. In the future, we need to expand the detailed phylogenetic analyses into the entire *L. chlorophorum* plastid proteome to examine the hypothesis based on the biosynthetic pathways for heme, Chl *a* and IPP.

## CONCLUSION

We here assessed the evolutionary origins of the proteins involved in the three plastid-localized pathways for the heme, Chl *a* and IPP biosyntheses in two separate dinoflagellate lineages bearing non-canonical plastids, namely kareniacean dinoflagellates bearing haptophyte-derived plastids (i.e., *Karenia brevis* and *Karlodinium veneficum*), and *L. chlorophorum* established a green alga-derived plastid. In each of the two dinoflagellate lineages, the three pathways have been modified differently during the process reducing an algal endosymbiont to a non-canonical plastid. We interpreted that the observed difference stemmed from the nature of the ancestral dinoflagellate engulfed a haptophyte/green algal endosymbiont. When individual pathways were compared between *L. chlorophorum* and *Karenia brevis*/*Karlodinium veneficum*, EGT appeared to contribute to the pathways in the former lineage much more substantially than those in the latter lineage. We proposed that this observation emerged partially from the structural difference in plastid-localizing signal (i.e., presence or absence of the SP) between the proteins acquired from the haptophyte endosymbiont and those from a green algal endosymbiont. The discussion based on the *Karenia brevis* and *Karlodinium veneficum* sequence data need to be reexamined in future studies incorporating the data from additional kareniacean species (e.g., members belonging to the genus *Takayama*), as well as their relative operating kleptoplastidy (*Gast et al., 2006*, *2007*).

## ACKNOWLEDGEMENTS

We thank Dr. Euki Yazaki (University of Tsukuba, Japan) for helping the phylogenetic analyses presented in this study. We also thank four reviewers for their constructive discussions and suggestions on this work.

### Funding

Eriko Matsuo was supported by a research fellowship from the Japanese Society for Promotion of Sciences (JSPS) for Young Scientists (no. 15J00821). This work was supported by grants from the JSPS awarded to Yuji Inagaki (23117006, 16H04826 and 17H03723). There was no additional external funding received for this study. The funders had no role in study design, data collection and analysis, decision to publish, or preparation of the manuscript.

### Grant Disclosures

The following grant information was disclosed by the authors:
Japanese Society for Promotion of Sciences (JSPS) for Young Scientists: 15J00821.
JSPS awarded to Yuji Inagaki: 23117006, 16H04826 and 17H03723.

### Competing Interests

The authors declare that they have no competing interests.

### Author Contributions

- Eriko Matsuo conceived and designed the experiments, performed the experiments, analyzed the data, contributed reagents/materials/analysis tools, prepared figures and/or tables, authored or reviewed drafts of the paper, approved the final draft, conducted phylogenetic analyses.
- Yuji Inagaki conceived and designed the experiments, performed the experiments, analyzed the data, contributed reagents/materials/analysis tools, prepared figures and/or tables, authored or reviewed drafts of the paper, approved the final draft.

### DNA Deposition

The following information was supplied regarding the deposition of DNA sequences:

The RNA-seq data obtained from a dinoflagellate *Lepidodoinium chlorophorum* was deposited in GenBank database under a DRA accession number DRA006544.

### Data Availability

The conceptional amino acid sequences are available from the Dryad Digital Repository: DOI 10.5061/dryad.591sm73.

### Supplemental Information

Supplemental information for this article can be found online at http://dx.doi.org/10.7717/peerj.5345#supplemental-information.

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
