# Peer review of "Patterns in evolutionary origins of heme, chlorophyll a and isopentenyl diphosphate biosynthetic pathways suggest non-photosynthetic periods prior to plastid replacements in dinoflagellates"

_PeerJ, doi:10.7717/peerj.5345_

## Round 0.1 · original submission · Minor Revisions

Your manuscript has been thoroughly revised by 4 people. Their comments have been very careful and constructive, and have helped me understand the validity of your findings. In my opinion, in line with reviewers, the manuscript is very interesting to a broad audience.
While there is a long list of corrections that need to be made, I still consider them as "Minor Revisions", because no new analysis are required (barring a minor request by reviewer #2).

I suggest that you take into consideration:

- some language correction

All reviewers indicated that while the manuscript is understandable, there are still several grammatical errors that require your attention.

- availability of data (alignments and trees)

Please make an effort to make all alignments and trees available.

- absence of UROS in KArenia /CPOX in Karlodinium

This is the only additional analysis suggested by reviewer #2. PLease take the comments into account and indicate whether these genes are indeed not present, and if so, whether thre could be a methodological bias.

- contamination from MMETSP or bioinfo pipeline

Again, please take reviewer #2 comments into account and clarify whether any contamination can be present.

- more detailed methodology for systematic reconstructions, more careful statements regarding tree interpretation

Two reviewers have indicated that they would like to have seen a more detailed explanation, and a more careful interpretation of trees.

- evolutionary history of Lepidodinium in relation to molecular dating

Reviewer #4 indicates an interesting aspect regarding the timing of endosymbiosis, which I think is worth thinking about.

I am certain that you will be able to address all issues, and expect a resubmission soon.

Best regards,

dan

·

Basic reporting

The literature on the topic is well cited. The figures effectively summarize the manuscript. Raw data are shared. The language is clear throughout most of the manuscript but there are parts where it needs to be improved. For a couple of sentences the grammar is clearly wrong.

This is largely an exploratory study, but it also serves to test one hypothesis, and provides a new hypothesis for one of the patterns found. Enough and sufficient backgrounds is provided for the reader to appreciate the relevance of the study and understand its contents.

Experimental design

This study presents original primary research that fits the aims and scopes of PeerJ. The research question is relevant and meaningful, but I wish the authors to better define it at the end of the introduction. As said before, this study is largely exploratory, but also tests a hypothesis. The methods are rigorous and appropriate to answer the research questions. The methods are mostly described with sufficient detail and information to replicate, but few comments on how to improve them can be found below.

Validity of the findings

The data are properly analyzed. The conclusions seem to follow the results logically. The hypotheses presented are interesting to stimulate further thought and can be linked to the patterns found.

Additional comments

In this manuscript, Matsuo and Inagaki investigate the phylogenetic provenance of the enzymes responsible for the biosynthesis of chlorophyll a, heme and isoprenoid precursors in dinoflagellates with derived non-canonical plastids (the fucoxanthin Karenia and Karlodinium, and the green Lepidodinium). Even though previous work had looked at the plastid proteomes in these species, no thorough attempt to compare all three pathways had been done before for all three species simultaneously. To achieve this, the authors provide a new full RNA-seq transcriptome, that of Lepidodinium, and perform phylogenetic analyses on each enzyme belonging to all three biosynthetic pathways. The goal of the authors is to find how chimeric the different plastid biosynthesis pathways in non-canonical dinoflagellate plastids are. In other words, are there patterns in the gene ancestry for these enzymes across the pathways in the different non-canonical plastids?

This is an interesting study. It is also well done. I think this study also helps to more systematically answer the question of what was the nature of the dinoflagellate hosts that acquired non-canonical plastids. It extends the hypothesis suggested for kareniaceans, i.e., that the host was heterotrophic with a relict plastid, to Lepidodinium. It seems to me that this is an interesting contribution because it suggests a convergent, and perhaps more generalizable, evolutionary route. It also implies that it is incorrect to call such examples ‘plastid replacements’ because both plastids co-existed at one point (and still might)—It was the acquisition of a new photosynthetic plastid alongside a no-longer photosynthetic relict plastid. This study also discovers interesting patterns of chimerism across these three biosynthetic pathways in plastids, and provides interesting hypotheses for the patterns observed, namely (1) none or few VI-type genes for the chlorophyll a biosynthesis pathway, and (2) few EA-type genes for all pathways in Lepidodinium.

I encourage publication of this manuscript in PeerJ. I do not have major objections to this study and its methods. My main comments concern the structuring of the manuscript.

My major comment is this:

- Why do the authors want to investigate the gene ancestries of precisely these three pathways? If their ultimate goal is to better understand chimerism in these plastid proteomes, it would then be best to look at the whole proteomes, or at least more pathways (e.g., fatty acid biosynthesis, Fe-S cluster biogenesis, etc.). But if their ultimate goal is to test the hypothesis of whether the host that acquired these non-canonical plastids was heterotrophic, then the experimental design seems more suitable. It does make sense that by comparing the ancestry of the chlorophyll a biosynthesis pathway (which is dispensable depending on lifestyle) to those for heme and isoprenoid precursors (which became essential to cells) such hypothesis can first be tested. At least three alternatives can be discriminated: (A) the host was heterotrophic with no plastid at all (predicts mostly EA-type and LA-type genes for all pathways), (B) the host was heterotrophic but with a relict plastid (predicts VI-type for heme and isoprenoid biosynthesis, but EA- or LA-type genes for chlorophyll a biosynthesis), and (C) the host was photosynthetic (predicts a significant number of VI-type genes for all three pathways). Such predictions rely on the safe assumption that genes are lost easily if their functions are not required. Alternative C has been largely rejected by previous studies. Other studies have also provided support for alternative B over alternative A. One of the main contributions of this study is that it lends further support to alternative B. I think that the reasoning behind the experimental design of this study can be made more explicit in the introduction.

I now provide specific comments on each section:

Title

- The title should be clearer and shorter. I think the major points made in the paper can be reflected in the title. For example: ‘Patterns in the origins of chlorophyll a, heme and isoprene biosynthesis pathways suggest a heterotrophic dinoflagellate host for non-canonical plastids.’ But other possibilities might be best.
- I also suggest using the word patterns rather than trends in the title and throughout the manuscript. A trend is a pattern throughout time. I think this study looks at patterns of gene ancestry among metabolic pathways.

Introduction

- I suggest removing the big section that describes the three pathways. They have been described elsewhere in the literature and make the introduction unnecessarily long. Fig. 1 is a good summary of such paragraphs. If desired, small molecule structures can be added to Fig. 1 to compensate for the lack of detail in the introduction after removing such paragraphs. This is all that is needed. I understand more background can help the non-expert, but a very long introduction and manuscript can also deter many other readers. We have all given up on manuscripts that become too long and tedious. One option is to move this text to the supplementary material. Mentioning the functionally equivalents (e.g., hemN and hemF, hemJ and hemY) of some of the enzymes can be kept in the introduction.
- Some of this content in the introduction becomes redundant with further explanations in the Results section about what the enzymes do, for example.
- Emphasize that you have sequenced a new transcriptome for Lepidodinium. This is valuable data.
- I am glad that the authors have chosen the word ‘chimeric’ to refer to the patterns observed. There has been an unfortunate trend in the literature in referring to such patterns as ‘mosaicism’. Mosaicism well refers to a combination of ancestral and derived characters, and not to a combination of mixed ancestries.

Methods

- Please make the alignments available online, especially because there were refined and trimmed manually which makes them less reproducible.
- Transcriptomes tend to be comprised by incomplete transcripts. Most often they are incomplete at the 3' end. This is also the end which carries a splice leader (SL) in dinoflagellates. But looking for SLs can add further support to the idea that these are transcripts are genuinely dinoflagellate and not contaminants.
- Please provide details about the sequencing of the Lepidodinium transcriptome, or cite a publication that does, and make it accessible to the scientific community.
- How complete is the Lepidodinium transcriptome? A BUSCO analysis, for example, can give a hint.
- Could there be LBA between some chlorarachniophyte and Lepidodinium genes? These could have been ‘green’ in origin, but similarly diverged considerably. Most sophisticated models could give clues, e.g., C10-60 models in IQ-TREE, or CAT-GTR in PhyloBayes.

Results

- The results section is organized around each biosynthesis pathway. This makes sense. However, the discussion provides too much detail. It can be summarized in some parts and refer to the Figs. 2-4 extensively. Please shorten. There appears to be some discussion in the results section too, e.g., 523-530.
- It is wise to withhold conclusions in the absence of statistical support for some of the trees, even though they might show some patterns in agreement with preconceived notions. I agree with the authors in their judgements.

Discussion

- Statement in 783-784 does not seem accurate. Please revise.
- Cavalier-Smith and Saldarriaga have also discussed such ideas (i.e., a hetrotrophic ancestor for non-canonical plastids in dinoflagellates) in their papers before. I suggest they are cited and discussed.
- The hypothesis put forward to explain why EA-type proteins are not common in these pathways in Lepidodinium makes sense. But the data are too limited. Looking at the whole proteome can more thoroughly test such an idea.

Figures

- The figures are well made and quite effectively summarize the manuscript. For example, Fig. 1 summarizes the biosynthetic pathways, Fig. 2-4 present simplified trees for all enzymes of the pathways under study, and Fig. 5 synthesizes the main findings in regard to the phylogenetic ancestries of each enzyme.

·

Basic reporting

The authors should ensure that all alignments and tree outputs generated in this study are publically accessible, either as supporting data here, or on an online repository, such as Treebase.

Experimental design

1) The absence of UROS/ CPOX from Karenia and Karlodinium respectively is puzzling, and I am not confident that this reflects, as stated, absence from the MMETSP transcriptome libraries. These libraries are massive- by my count about 72000 non-redundant transcript sequences in the Karlodinium combined assemblies, and over 400000 between all four Karenia combined assemblies, and in each case there are four independent replicates of each transcriptome library- I fail to see how a fundamental component of tetrapyrrole synthesis would not be detected in these libraries, particularly if all other components are. Other possibilities that should/ could be considered:

- Is there any evidence for UROS transcripts in the unassembled reads (as opposed to assembled transcripts) from Karenia, or likewise for CPOX in Karlodinium?
- Are there alternative pathways that might function to convert hydroxymethylbillane into protoporphyin IX in either species? How many different isoforms of each sequence did the authors use as search queries? Might (for example) a mitochondria-targeted protein, or query from a plastid-lacking species, give different results than a query of plastid/ cyanobacterial origin?
- Is there any possibility that either UROS or CPOX has become fused to another protein in the tetrapyrrole biosynthesis pathway in either species- e.g. is there a UROS domain on Karlodinium HMBS or similar?

2) The only other major concern I have is of contamination in some of the transcriptome libraries considered, which may either result from non-axenic cultures or (in the case of MMETSP) technical contamination resulting from pooling libraries from different species in the same sequencing reaction. Ideally the authors should use decontaminated versions of the Karenia and Karlodinium reference libraries (c.f. Marron et al., 2016; Dorrell, Gile et al., 2017) to avoid mis-identifying sequences of non-dinoflagellate origin as being present in each species.

Furthermore, the authors should provide evidence that any transcripts with atypical evolutionary inheritance patterns (and particularly transcripts for which two orthologues of separate evolutionary origin, c.f. the prokaryotic red UROS, and GTR, are detected), are indeed genuine components of the transcriptomes in question: this could be a matter of identifying the same transcript repeatedly in multiple libraries from the same species, or for example looking at the nucleotide composition and/ or codon usage patterns associated with the atypically inherited transcripts. The phylogenies here do not need to be redone, but the authors should be prepared to justify that isolated and/ or atypically inherited transcripts identified in each species are genuine.

Validity of the findings

Some of the evolutionary assignations made by the authors are questionable. For example, lines 352-353 state "The second Karlodinium [UROS] sequence… tied with the sequence of a red alga Rhodosorus marinus … suggesting that the Karlodinium-2 sequence was acquired from a red alga". Consulting the tree topology shown in fig. S1, I see no convincing evidence for a red-to-Karlodinium transfer: rather the Karlodinium/ Rhodosorus clade form an isolated group within a clade of otherwise prokaryotic sequences, and indeed all other red algal sequences in the tree (including one other isoform from Rhodosorus). It is equally possible in this case that the Karlodinium sequence was acquired from a bacterium, then transferred to Rhodosorus; that both lineages acquired orthologous sequences from related bacteria; or that either of both sequences are the result of prokaryotic contamination in the Rhodosorus/ Karlodinium sequence libraries (see above).

As a general rule of thumb, I would only recommend assigning specific evolutionary origins if the Karenia/ Karlodinium sequences resolve within a paraphyletic group of organisms with a common evolutionary origin, or at the very least that there are a minimum of two or three sequences with the sister-group affiliation in the tree. This may reduce the number of (particularly LA-) inferences made in the manuscript, but I would rather the authors present a smaller number of more credible relationships, than try to report everything that may or may not be present.

Additional comments

General manuscript point: full species names need to be supplied throughout- there are more than one Karenia (brevis, mikimotoi), Karlodinium (veneficum/ micrum, armiger) and Lepidodinium species (chlorophorum, viride) that exist!

Line 59: should be "the vast majority"

Line 78-83: it would be good to mention recent work (c.f. Matsuda et al., 2017, Zerdoner Calasan et al. 2017) indicating that dinotom endosymbionts are evolutionarily transient, potentially subject to frequent replacement, and therefore a direct comparison between them and more evolutionarily permanent plastid lineages is inappropriate

Line 84: should be "as prey". Could you give some examples of key genera, and the types of algae they ingest?

Lines 89-96: I would like to see a slightly more extensive discussion of what is already known about host-derived contributions to fucoxanthin and Lepidodinium plastid proteomes- as it is greater than is suggested by the authors. In particular, they should acknowledge (either here, or in the Discussion, lines 807-809; 846-849):

- There are actually quite a lot of examples known of proteins, targeted to serially acquired dinoflagellate plastids, of peridinin origin- alongside Minge 2010 and Bentlage 2016, the authors could/ should cite Patron 2006; Nosenko 2006; or Dorrell and Howe 2015 (as a summary of everything that has previously been identified to be of peridinin origin in each lineage)

- Among the proteins identified to be of peridinin origin are subunits of the photosynthetic electron transport machinery (e.g. atpC in Karlodinium; Patron et al., 2006; and FNR in Lepidodinium, Minge et al., 2010). This argues against (lines 807-809. 846-849) the common ancestors of the fucoxanthin and Lepidodinium lineages being obligately non-photosynthetic, or at the very least implies that whatever loss of photosynthesis occurred was relatively transient, and did not lead to the loss of all photosystem-associated genes of peridinin origin from the host genome.

- Similarly, the presence of RNA modification pathways derived from the ancestral peridinin plastid in the fucoxanthin plastid lineage (c.f. Dorrell and Howe, 2012; Jackson et al., 2013; Richardson et al., 2014), might be consistent with the ancestor of the fucoxanthin lineage retaining some form of plastid genome, even if this genome was associated with a leucoplast or vestigial plastid, from the peridinin lineage, until shortly before the acquisition of the replacement. The authors could also note that peridinin-type RNA modificiation pathways do not function in Lepidodinium (Richardson et al., 2014), which mirrors the lower incidence of EA-derived proteins in heme and IPP synthesis.

Line 102: should be "a sppA gene, encoding serine proteease IV, of green algal origin was detected in Karenia…and a csp41 gene, encoding an mRNA-binding protein, of putative stramenopile origin was detected in Lepidodinium". How robust is the support for these relationships?

Line 108: should be "varies"

Line 109-111: "Likewise, we are uncertain…plastid" doesn't make sense. Perhaps the authors mean "we are uncertain whether there is an evolutionary consistent trend in chimerism"- please check and verify

Line 126: should be "We are aware"

Line 132: should be "is catalysed"

Line 148: should be "use the pathway"

Line 149-151: rephrase. Typically the past tense is only used when referring to a specific study or time point (e.g. "In a recent study by Koreny et al., 2013, neither X nor Y were detected")- otherwise this should be in the present tense (e.g. "Neither X nor Y appear to be present")

Lines 152-154: more strictly chlorophyll and heme share a common precusor biosynthesis pathway, which branches based on the chelation of either Mg/ Fe and the downstream processing steps that occur

Line 159-166; lines 580-582: to my knowledge MgPME cyclase has never been detected in eukaryotes outside of the land plants (c.f. Wilhelm et al., 2006; Nymark et al., 2009), and ycf59 (the inferred membrane-bound subunit of MgPME cyclase cf. Peter et al., 2011) typically has a restricted, plastid-encoded distribution outside of the green lineage. So I would be cautious in general about emphasing the presence, origin, or functional importance of MgPME cyclase, at least until we have a better idea of the diversity of chl branch synthesis pathways in non-green plastids.

Lines 239-241: did the authors look at the individual transcriptome libraries, or combined assemblies, generated for Karenia and Karlodinium? This is particularly important in the context of contamination (contaminant sequences may not be present in all libraries from one species), but in either case the specific version and accession date of the transcriptome libraries should be provided.

Lines 242-243: the final set of query sequences used for this study should be presented as a supporting dataset

Line 301-302: to my knowledge, SignalP 4.1 does not allow the prediction of TP-like regions- do the authors mean TargetP1.1 (for example), applied to a truncated sequence following prediction of an SP by SignalP4.1? If so, clarify. It would also be good to briefly profile the length and net charge associated with the predicted TP regions in each sequence, given that previously characterised nucleus-encoded and plastid-targeted proteins in fucoxanthin-containing dinoflagellates are demarcated by having a relaxed charge balance (Patron and Waller, 2007) and longer TP region (Yokoyama et al., 2011) than equivalent BTSs from haptophytes.

Line 336-337, and elsewhere: if the BLAST analyses were performed using the individual libraries (rather than pooled assemblies) from Karenia and Karlodinium, it is possible that monophyletic groups of sequences from one species might in fact be different assemblies/ annotations of the same transcript from different libraries, as opposed to different transcripts from paralogous genes. At least one of the Karlodinium PGBD sequences, for example, is on a very long branch, which could be consistent with a partial/ incomplete copy of the same gene, as opposed to divergent paralogues, from different transcriptome libraries. Alignments in these cases would be useful, particularly to identify base polymorphisms or other features that distinguish different transcripts from the same species, and are unlikely to be the result of assembly.

Line 392: should be "a single version was identified in Karlodinium"

Lines 423-425: "pioneering studies have documented such evolutionarily chimeric plastid proteomes in dinoflagellates bearing non-canonical plastids, but primarily emphasized the presence of EA-type proteins to demonstrate the integration of a haptophyte endosymbiont into the dinoflagellate cell as the plastid." - citations needed here.

Line 553: chlI has in fact been detected in plastid transcriptome data from Karenia mikimotoi (Dorrell et al., 2016). It would be useful to inspect this sequence for the presence of a 3' poly(U) tail, which is characteristic of dinoflagellate plastid-derived transcripts in MMETSP data (Dorrell, Klinger et al., 2017)

Lines 600-601: technically it is possible that there is a light-independent POR in the Karlodinium plastid genome as the published genome sequence is incomplete, although no such transcripts are detected in Karenia mikimotoi plastid transcriptome data (as above)

Line 796: should be "in heme biosynthesis"

Line 806: should be "notably"

Lines 832-834: the authors could mention here that Karenia and Karlodinium appear to be quite distantly related to one another (Bergholtz et al., 2006), and indeed their plastid genomes have different associated gene contents (Gabrielsen et al., 2011; Dorrell et al., 2016), which would be consistent with their assertion of divergent evolution in plastid proteome content between the two genera.

Lines 857-876: The authors discuss possible factors underpinning the presence of laterally acquired proteins, such as the presence of chlorarachniophyte-derived proteins in the chl biosynthesis branch of Lepidodinium. While this discussion is detailed, I think that it misses the possibility that the LA-derived proteins might have originated from an endosymbiont that was already chimeric in nature, prior to uptake. This is particularly likely for the fucoxanthin plastid, as its ancestor occupies a very deep position on the haptophyte tree; as a sister-group to the prymnesiophytes, and we have no real idea what suite of plastid-targeted proteins it possessed prior to uptake. Likewise it would be interesting to consider whether there may have been horizontal gene transfer from chlorarachniophytes into the pedinophyte ancestor of the Lepidodinium plastid, particularly given the presence of an endosymbiont-derived POR in the Lepidodinium transcriptome.

Line 909: should be "much more substantially"

·

Basic reporting

The article is in most cases clearly written and in general easy to read. However there are a number of small mistakes throughout the manuscript (wrong usage of tenses and several words, is/are or was/were mistakes, missing “the”s and some grammatical issues) as well as several sentences that need rewriting (e.g. page 11, lines 5-8; page 11, lines 20-23; page 17, line 5-7; page 27, lines 13-15; page 41, lines 20-23). Therefore I suggest that the authors ask a proficient English speaker to go through the manuscript and correct those small mistakes, which would improve readability and make for an easier understanding of the article.

The authors include literature references throughout the article and seem to be up-to-date within the field (except for a couple issues I am going to mention below in “Validity of the findings”). There are a few statements where references should be added:
Page 6, lines 19-23:
The three pathways in photosynthetic eukaryotes and those in cyanobacteria are principally homologous to each other, suggesting that the pathways in photosynthetic eukaryotes can be traced back to those endosymbiotically acquired from the cyanobacterial endosymbiont that gave rise to the first plastid (primary plastid).
Page 8, lines 2-6:
Overall, land plants used the pathway of cyanobacterial origin with modifications described below. (i) PBGD, UROS, and UROD, which are likely acquired from phylogenetically diverse organisms in non-endosymbiotic contexts, were identified. (ii) Neither hemN-version of CPOX nor hemJ-version of PPOX was detected.

The authors provide sufficient background information for their research and one might argue – and this is a minor quibble - it is almost a bit too detailed. All three pathways discussed are well described in the literature and while it is understandable that the authors want the information for the reader to be readily available, the pathway description is already extremely detailed in the Introduction and then the function of the single enzymes is described again in the Results. Altogether this makes for a very long read and I think the article would benefit from making at least the Results section a bit more concise (i.e. not repeat parts from the Introduction to such an extent). A good example is the subchapter “Genetic influx from phylogenetically diverse organisms shaped the Lepidodinium pathway”, which is more concisely written than the rest and still clearly conveys the results.

Experimental design

The authors state that their goal is a systematic evaluation of the impact of “exogenous” genes on several metabolic pathways situated in non-canonical plastids of dinoflagellates. Such a systematic analysis and comparison between non-canonical dinoflagellate plastids of different origins has not been undertaken before and provides valuable insight into the evolution of such plastids.
The tree reconstructions were performed with two different methods, with the maximum likelihood trees having been calculated with the best fitting model for each individual alignment and the Bayesian analyses being based on four independent chains, indicating a careful and concise methodological approach by the authors.
Minor comments:
- it would be helpful for readers to add a list of all taxa used in the analysis as Supp. Table
- what do the dashed branches in the tree figures represent?
- the algorithm(s) used with MAFFT should be indicated
- define KOIDs
- indicate which KOIDs belong to which pathway
- it is not clear which sequences from the KEGG database were used as queries in the initial BLAST - all of them? (see page 12, lines 8-9)

Validity of the findings

Overall the findings are represented in a clear and valid manner, however I do not agree with several concluding statements which I will detail below:

A. Several times in the Results section the authors claim a phylogenetic association that is not very well supported by the trees in question. In general, those are phylogenies where the query sequence clusters with only one sequence from another eukaryotic group and in one case this group is only represented by this one sequence in the tree. Examples are:

- in the UROS tree (Karlodinium-2 clusters with a single red algal sequence; page 17, lines 8-11)
- in the UROD tree (Karenia-3 clusters with the only euglenozoan representative in the tree; page 18, lines 1-3)
- in the ispD tree (Lepidodinium-1 clusters with a single stramenopile sequence in the absence of a monophyletic, well-supported stramenopile clade; page 33, lines 5-8)

I think in those cases the authors should make more careful statements and include other possibilities leading to such topologies, such as low taxon sampling (euglenozoa), relatively short sequences (single gene trees) and contaminations (MMETSP data are known to be contaminated), into their conclusions.
Additionally, the statement on page 34, lines 23-25 is not supported by the tree – the topology placing the apicomplexans/chromerids within the dinoflagellates is not supported and the authors state themselves that the removal of the rapidly evolving apicomplexan/chromerid sequences strongly improves the support of the clade. It should be re-formulated as well.

B. The authors base some of their conclusions (see also point C below) on the wrong assumption that Karenia and Karlodinium share the same endosymbiont, i.e. the ancestor of both has already acquired the endosymbiont. While it cannot be excluded that the ancestor already had a symbiont before the divergence, the current haptophyte plastids in those two lineages come from two different sources as shown by Gast et al., 2007 (doi:10.1111/j.1462-2920.2006.01109.x) in a 16S phylogeny. Therefore, the authors should rephrase the fragment on page 29, lines 8-9

“which were acquired from the haptophyte endosymbiont resided in the ancestral kareniacean species”

and reformulate the statement in their Discussion on page 38, lines 14-16:

“These observations suggest that the reconstruction of metabolic pathways in the haptophyte derived plastids (i.e. gene acquisitions/losses) was not completed before the separation of the genera Karenia and Karlodinium.”

C. A minor issue but although there is no Karenia plastid genome, there is a Karenia transcriptome available, created by Dorrell et al., 2016 (https://doi.org/10.1007/s11103-015-0408-9), that allows comparison of the gene content of both plastids (which the authors of that publication do). According to that publication, Karenia plastids encode ChlI, like the Karlodinium plastid, but their overall gene content is not the same, so the authors should rephrase the following statements and update their references:

Page 26, lines 3-7:
“Although no plastid genome data is available for Karenia, we identified the ChlI sequence in the transcriptomic data of this species (contig No., 0173787962). According to the intimate organismal relationship between Karlodinium and Karenia, we considered that the Karenia ChlI sequence was transcribed from the plastid genome.”

Page 28, lines 8-10:
“No gene for light-independent POR was found in the plastid genome of Karlodinium (Gabrielsen et al. 2011), implying that kareniacean species lack the light-independent version.”

D. A very minor issue, but the authors should mention the difference in results between their GSAT analysis and the phylogeny recovered by Cihlář & Füssy et al, 2016, as they specifically discuss Karenia in that context.

·

Basic reporting

The article is written in professional English, however, it countains several grammatic and stylistic mistakes, a few sentences are missing words (e.g., lines 83, 126, 253, 326) and would benefit from proof reading by a native speaker. The wording is sometimes a bit clumsy, but the text is unabiguous and clearly understandable. There are a few typographical mistakes such as combining "twenty two" and "27" in the same sentence (line 543).
The literature is properly referenced and the sources are relevant and numerous.
The introduction provides good context and successfully explains the relevance of the study and that the evolutionary story it wants to deliver is interesting. The authors manage to clearly describe the aims of the study including a good clarifying disclaimers on which topics will not be addressed and why. The Introduction chapter contains quite detailed description of the metabolic pathways and their particular steps which I find a bit unnecesarilly extensive and detailed since the paper is of evolutionary biology, not biochemistry. The text on lines 211-225 should be a part of Results or shortened significanlty to form a brief conclusion for the Introduction and outset for the next chapters describing the results.
The title of the paper is descriptive and clear, I would however suggest a shorter title and omit the unnecessary buzzword ("Trends in X").
I see a problem with structuring of the whole article. First, the distincion between "Results" and "Discussion" sections is not fully respected: the paper reads kind of like it was originally written in a format that merges "Results and Discussion" into one chapter. The "Results" section contains many interpretations of the presented data and speculations made based on them (e.g., lines 369, 387, 408, 411, 490, 507, 523, 542, 572, 623, 756 and many more), which I believe should be a part of "Discussion". The chapter is also very long and sometimes exhausting to read: ommiting the interpretative parts and structuring the text into smaller subchapters (I would suggest a subsection for each gene, and would prefer the Kareniacean and Lepidodinium phylogenetic data to be adressed together as they are based on the same trees anyway.) The "Discussion", on the other hand, is quite brief, albeit poignant and telling an interesting story. I have several suggestions for additional questions to discuss (see the end of the “Validity of findings“ section).
The text contains quite a lot of abbreviations. They are in most cases clearly explained at their first occurence, but nevertheless, I would find the inclusion of an index of used abbreviations useful.
The article includes five color figures which are well labeled and of good esthetic quality.
Figures 2-4 are nice and clear and elegant to look at but would lose informativity if printed by black and white printer or read by a person with some degree of colorblindness (around 7 % of males). I would strongly suggest adding another distinction between the colored taxons (symbols, abbreviations or line type). Moreover, the color distinctions are described in the text and might be a bit confusing since the verbal description of the colors might differ subjectively and between readers from different language/cultural backgrounds - the legends included directly in the figures would solve this issue. On the other hand, figure 5 would benefit from color distinction between the four categories (for example red, i.e. same as in the trees, for VI, brown/orange for haptophyte EA and green for chlorophyte EA, another color for LA, and grey for unknown), the differencies in the ratio of these categories in the two groups of organism would be more visible.
A few requests for clarifications:
- On line 82, authors suggest that the interlock between host and endosymbiont naturally lead to their genetic connection. This doesn't need to be the case since there are endosymbioses known to exist for hundreds of millions of years without its members merging into a genetic chimera.
- On line 118, the authors state that the [metabolic pathays] are "nucleus-encoded" but do not clarify which organisms/taxons are they refering to.
- On line 600, the authors state that no light-independent POR was found in the previous analysis of Karlodinium transcriptome. Does this mean they also searched for it in their data and failed?

Experimental design

I find the experimetal design to be appropriate to the aims of the study.
The investigation was performed to appropriate methodological and technical standards

The supplemental data contain full trees but not the alignments used for their construction which unfortunately makes the results non-reproducible, especially since the editing/trimming of the alignments was done manually. Please, include the alignments and ideally a file with the exact parameters of the RAxML analysis.
Some informations are present in the supplemental files, but they are not references in the text (the method for transit peptide prediction, substitution models selected by ProtTest...).

Validity of the findings

The study presents novel findings on the evolutionary stories behind a wide set of plastid metabolism-related genes in three species of dinoflagellates with non-peridinin plastids, many of which were first identified in the sequence data in terms of this study.
Data is robust and statistically sound as much as an evolutionary study can be. The phylogenetic analyses were done responsibly and the interpretations of the tree topologies is appropriate: the authors infer evolutionary scenarios based on the high-supported data only, and treat the low-supported results with corresponding caution.
The authors use the profile of gene origins (vertical inheritance, endosymbiotic transfer or lateral transfer) to reconstruct the evolutionary history which led to the establishment of the extant plastids in Karenia, Karlodinium and Lepidodinium which replaced the ancestral peridinin-containing plastids. The Discussion addressing these scenarios is performed with caution, yet it still postulates several intriguing stories and bright, out-of-the-box ideas.
Here are a few suggestions/questions to include in the Discussion to make it more rich:
- The authors decided to classify the gene origins as "vertical", "endosymbiotic" and "lateral". The "lateral" category includes both plastid-containing organism and organisms without plastids. What is the author's reasoning behind not separating these two subcategories?
- The authors keep refering to dinoflagellate genes related to/nested inside chlorarachniophytes as "acquired from chlorarachniophytes". Is there any taxonomical or evolutionary datation evidence for this claim? As far as I know there is no datation available for any of the endosymbiotic events in question - so the direction of the transfer cannot be inferred with certainty.
- How would the relationship between the dinoflagellate and the chlorarachniophyte donor of plastid-related genes look like? Are there any hints in the ecology of the extant species? Where do organisms of these taxons live, do their geograophical/ecological niches overlap? What would drive the frequent shifts between predation and phototrophy?
- Is it possible that the peridinin-containing plastid and the haptophyte-related plastid coexisted for a certain period of time in the dinoflagellate cell? How would this reflect on the gene phylogenies?
- Do I understand it correctly that the authors postulate two cycles of plastid photosynthesis loss and subsequent establishment of a new, unrelated one, for the evolutionary history of Lepidodinium on lines 846-849? Does this scenario makes sense in regard to the datation / the millions of years needed to go throught such changes? Any estimate on how old the plastids are? Where would the selective pressure to switch so frequently between non-photosynthetic plastid / predation / phototrophy come from? Which enviromental or ecological changes would cause such frequent shifts in the selective pressure in one lineage?

---

## Round 0.2 · Minor Revisions

Thank you for your revised paper. There are now only minor issues that need to be corrected, which I expect will be fairly simple (mostly pointed out by reviewer #2)

·

Basic reporting

Line 90: should be “are composed of proteins with diverse evolutionary origins”
Line 110: should the “their cyanobacteria ancestry”
Line 129: should be “neither of the enzymes…has been detected in complex algae”
Line 306: should be “hydroxymethylbillane”
Line 425: should be “the donor of the GTR gene to L. chlorophorum remains unclear”
Line 502: should be “in heme biosynthesis”
Line 505: should be “is most prominent”
Lines 705-706: should be “two scenarios remains possible” and “originated from a stramenopile”
Line 723-724: should be “recovered a monophyletic clade containing the two…”

Experimental design

For lines 532-534: it would take all of about 3 days to verify the presence of a poly(U) tail on the K. brevis chlI by RT-PCR with an oligo-d(A) primer, if the authors had RNA/ cDNA/ time remaining to do this. I stress this is completely optional, but it would be fast and easy to perform.

Validity of the findings

thank the authors for exercising caution in defining the phylogenetic origins of LA-type genes with uncertain topology. I stress that this was requested not due to discomfort over the idea of LA-type genes, which is a really nice idea. I made these requests purely to maximise the subsequent impact and credibility of this study, i.e. to avoid giving ammunition to peer researchers sceptical of HGT through the over-interpretation of ambiguous tree topologies. By minimising this the authors also draw attention to very well-resolved examples of LA genes in their data, e.g. the green algal ChlH genes in Kareniaceae.

The only remaining statements where I could, in theory, drive a nail would be-

Lines 473-474: it isn’t clear directly from Fig. 2G that the Lepidodinium-2 and -3 CPOX sequences are specifically of diatom origin, rather they form a sister-group to a clade of diatoms, with Chromera as a sister-group, followed by Chattonella as an outgroup. The authors don’t need to revise the figure in any way, but I would recommend downgrading these enzymes to being “of stramenopile origin” or “ochrophyte origin” for posterity. Conversely the Lepidodinium ChlD could be upgraded to being specifically of diatom origin based on the clade composition (lines 645-646).

Lines 579-580: the bootstrap support for all of the sub-clades within the haptophyte/ stramenopile clade are too weak to be certain of an LA v EA origin. I would suggest either dropping this statement or provide the alternative that this could be an EA gene with a poorly composed clade. Alternatively, if sister-group identity (rather than bootstrap resolution) were taken as adequate for defining evolutionary origin, then presumably the Clade 1 POR in both Kareniaceae (lines 594-595) should presumably be EA rather than ambiguous EA/LA.

Lines 705-706: presumably then Ochromonas could also have picked up an IspD gene from Lepidodinium. Perhaps it would be better to say that “the IspD sequence was transferred between Lepidodinium and Ochromonas”, which makes no assertion of direction.

Lines 847-849: while the Gast et al. study does show a distinct origin for Karenia and Karlodinium, the bootstrap support is non-existent, and a lot of prior and subsequent multigene studies (c.f. Takishita et al. 1999, 2000, 2005; Gabrielsen et al. 2011, Dorrell and Howe 2012) resolve monophyly for the fucoxanthin lineage. I would recommend not discussing this point. I do agree that getting more haptophyte sampling depth would be ideal to resolve the precise evolutionary histories of the LA proteins identified in this study.

Additional comments

I am largely happy with the revisions made to this manuscript. Provided the attached revisions are made, I feel it is ready for publication.

·

Basic reporting

The article is written in a clear, professional English, although it is still not perfect from the stylistic and grammatical standpoint. The sources are properly referenced and up to date, with a number of them being added recently to the paper's advantage. The introduction has been significantly shortened and made more poignant and easier to read, it communicates the context and aims of the study well and provides clear outline of what the authors want (and do not want) to address and investigate and why. The study discusses an impressive number of separate phylogenetic analyses which makes the resulting descriptions of results and discussion quite long and exhausive, and makes it hard and even contra-productive to try to clearly separate "results" from "discussion". The authors, nevertheless, did their best to present their findings in an organized and pregnant way. The text is now a bit briefer and no longer contains redundant passages. The requests for clarification or re-wording I posed previosuly were answered to my satisfaction.

Experimental design

The experimetal design is appropriate to the aims of the study which are clearly described and substanciated. The investigation was performed to appropriate methodological standard and the authors generated great amount of data and phylogenetic analyses. The requests for alignments and elaboration on some of the methodic approaches were answered and the information in question is available in either supplemental data or the text itself.

Validity of the findings

The study presents an evolutionary story behind a relatively wide set of plastid metabolism-related genes in three species of dinoflagellates whose ancestral peridinin plastids were lost ad replaced in the course of evolution. The set of genes/metabolic pathways was selected quite arbitrarily - as a model to investigate more genral trends in the dinoflagellate plastid evolution, this approach is made clear by the authors and justified. In fact, the volume of analyzed data (the number of observed genes times the number of organisms with different plastid evolutionary history) is very impressive, especially considering this is a study by two authors. Many of the genes were first identified or first sequenced in terms of this study.

Data is robust and statistically sound as much as an evolutionary study can be. The phylogenetic analyses were done responsibly - the authors used two independent methods and each alignment was assigned the corresponding substitution model individually. All the used data was also thoroughly screened for possible contaminations. The interpretations of the tree topologies were done with corresponding caution, yet they still postulate several interesting stories and clever, out-of-the-box ideas.

Most of the suggestions for the discussion enrichment I posed previously were not included in the manuscript, but I appreciate the authors effort to keep the text shorter and "down to Earth",i.e. devoid of too much of a theorizing.

---

## Round 0.3 · accepted · Accept

I believe the author's have addressed all reviewer comments appropriately, to the best of their ability and keeping in mind the focus of this paper.

I'd like to thank both the authors and reviewers for such a thorough peer reviewing process.